# GBIR: A NOVEL GAUSSIAN ITERATIVE METHOD FOR MEDICAL IMAGE RECONSTRUCTION

## ABSTRACT

Computed Tomography (CT) and Magnetic Resonance Imaging (MRI) are crucial diagnostic tools, but undersampling techniques like Sparse-View CT (SV-CT) and Compressed-Sensing MRI (CS-MRI), aimed at reducing patient exposure and scan time, make image reconstruction more challenging. While deep learning-based reconstruction (DLR) methods have made significant strides, they face limitations in adapting to varying scan geometries and handling diverse patient data, hindering widespread clinical use. In this paper, we propose a novel **G**aussian-**B**ased **I**terative **R**econstruction (**GBIR**) framework that uses learnable Gaussians representations for personalized medical image reconstruction, addressing the shortcomings of DLR methods. GBIR optimizes case-specific parameters in an end-to-end fashion, enabling better generalization and flexibility under sparse measurements. Additionally, we introduce the **M**ulti-**O**rgan Medical Image **RE**construction (**MORE**) dataset, comprising over 70,000 CT and MRI slices across multiple body parts and conditions. Our experiments show that GBIR outperforms state-of-the-art methods in both accuracy and speed, offering a robust solution for personalized medical image reconstruction.

## 1 INTRODUCTION

Computed Tomography (CT) (Koetzier et al., 2023) and Magnetic Resonance Imaging (MRI) (Harisinghani et al., 2019) are the two most important diagnostic technologies in modern medicine. CT scans use computer processing to reconstruct detailed cross-sectional images from X-rays emitted at various angles and measured as they pass through body tissues. MRI scans use powerful magnets and radio waves to excite hydrogen atoms in the body, generating signals that are detected and processed by a computer to create detailed images of internal structures. Therefore, sophisticated image reconstruction algorithms are essential for both CT and MRI, converting raw data from multiple projections into diagnostic images (Szczykutowicz et al., 2022; Zhu et al., 2018). Modern medical practices use undersampled raw measurements by reducing radiation exposure or scanning time to benefit the health and improve comfort of patients, for example, adopting Sparse-View CT (SV-CT) (Koetzier et al., 2023) and Compressed-Sensing MRI (CS-MRI) (Lustig et al., 2008), as shown in Figure 1. However, these undersampling procedures make the reconstruction process much more challenging, as the raw measurements are insufficient to recover the true 3D conditions within the patient's body.

The scanning process by the machine is usually called the forward process, which acquires the raw measurements from the patient. Conversely, the reconstruction process is called the inverse process that recovers the 3D volume from the raw measurements. The forward process is well studied and can be modeled by mathematical equations, but the inverse process is actually an ill-posed problem with non-unique solutions that is challenging to solve. Traditional methods for medical image reconstruction, such as Filtered Back Projection (FBP) (Bracewell & Riddle, 1967) and Inverse Fast Fourier Transform (IFFT) (Gallagher et al., 2008) for CT and MRI, are incapable of handling the reconstruction problem from sparse measurements. Deep learning-based reconstruction (DLR) methods are leading advancements in medical image reconstruction, offering practical solutions such as SV-CT and CS-MRI for medical diagnosis. While various types of DLR methods exist, such as direct learning methods (Zhu et al., 2018; He et al., 2020), image-domain denoising methods (Jin et al., 2017; Chen et al., 2017), and dual-domain reconstruction methods (Hu et al., 2020), they all share a common principle: employing neural networks to learn the mapping from the measurement domain to the image domain. Nevertheless, SV-CT and CS-MRI have seen limited adoption in

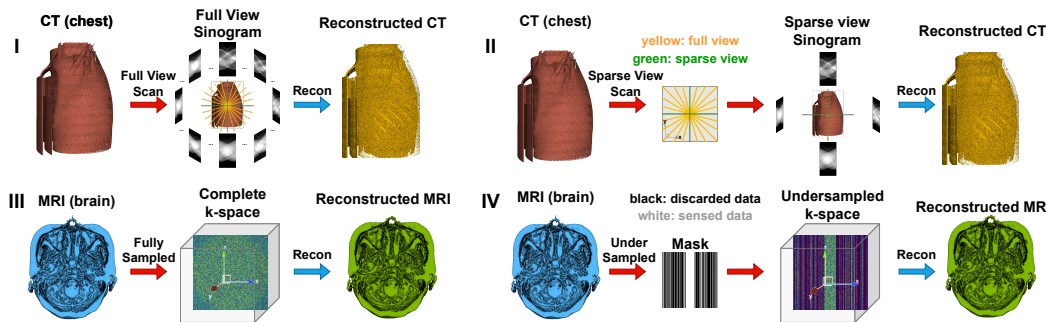

Figure 1: Illustration of medical image reconstruction paradigms. I: Full-View CT scans the patient from multiple angles to acquire complete measurements.; II: Sparse-View CT reduces the number of views to reduce radiation exposure.; III: Complete MRI captures full data sets for high-resolution imaging, ensuring detailed anatomical visualization. IV: Compressed-Sensing MRI reconstructs images from undersampled data, significantly reducing scan time.

clinical practice (Koetzier et al., 2023; Jaspan et al., 2015). The underlying reason is the inherent limitations of neural networks. Firstly, the fixed mapping learned by neural networks poses challenges in adapting to varying scan geometries. For instance, an SV-CT model trained on 60 views cannot be easily extended to 120 or 180 views without undergoing a complete retraining process. Secondly, the effectiveness of neural networks is limited by the diversity of the training data. Variations in patient demographics and medical conditions make it hard to create a comprehensive dataset. Consequently, DLR methods may struggle in clinical practice, as neural networks might fail to reconstruct images for conditions not included in the training data. As noted by Szczykutowicz et al. (2022), future methods should be customized for each individual patient.

Given the numerous inherent limitations of neural networks in medical image reconstruction, we are motivated to take a bold step: abandoning neural networks in favor of a set of learnable isotropic Gaussians to represent the 3D volume to be reconstructed. This idea is inspired by the success of 3D Gaussian Splatting (3DGS) in the field of computer graphics (Kerbl et al., 2023), which uses a set of 3D Gaussians to represent and reconstruct a 3D scene from 2D images. But it is important to note that, unlike 3D scene reconstruction, medical image reconstruction involves supervision signals in the measurement domain rather than the image domain, and the objective is to recover a fixed 3D volume instead of rendering a dynamic 3D scene. Without any Rendering process, in this paper, we propose a novel **G**aussian-**B**ased **I**terative **R**econstruction (**GBIR**) framework that encompasses both high-quality representation and an efficient reconstruction process. GBIR creates a tailored Gaussian representation for each case (patient), with learnable parameters optimized in an end-to-end fashion. This allows for customized medical image reconstruction, overcoming the generalization challenges faced by neural networks, and it also offers flexibility in reconstructing medical images under varying sparse measurement conditions. GBIR requires only the current patient's data for optimization, enabling a "train-as-you-infer" approach.

The main contributions of this paper can be summarized as follows:

- We propose a novel Gaussian-Based Iterative Reconstruction (GBIR) framework. GBIR employs a new reconstruction approach that involves projecting onto the measurement at each iteration, and optimizing the reconstruction based on the loss with the current case's measurement. This method achieves personalized modeling and strong generalization.
- We propose a comprehensive **M**ulti-**O**rgan Medical Image **RE**construction (**MORE**) dataset, which contains over 70,000 slices from 173 patients, covering 15 body parts in CT scans and 5 body parts in MRI scans, with various types of diseases. The dataset has passed the ethical review of the hospital and the local ethics committee and will be released to the public.
- We conduct extensive experiments to evaluate the performance of our proposed method, we compare GBIR with various existing methods on the proposed MORE dataset and other public datasets. The results show GBIR achieves state-of-the-art performance, outperforming other baselines by an obvious margin, and demonstrates superior inference speed.

## 2 BACKGROUND

**Problem Definition** The forward process in medical imaging systems (*e.g.*, CT, MRI) can be formulated as follows:

$$y = \mathbf{A}x + n, \tag{1}$$

where $x$ is the 3D volume of the patient, $A$ is the system matrix that models the imaging system, $y$ represents the acquired measurements, and $n$ is the noise. Medical image reconstruction refers to the inverse problem of recovering the 3D volume $x$ from the measurements $y$. In applications like SV-CT and CS-MRI, the matrix $\mathbf{A}$ is sparse and the measurements $y$ are undersampled, increasing the complexity of the reconstruction process. This inverse problem is inherently ill-posed and non-unique, with the goal being to estimate the most likely 3D volume that corresponds to the given measurements.

A typical approach involves minimizing a loss function that balances the fidelity to the measurements $y$ and the regularization term that imposes prior knowledge about the structure of $x$. The optimization problem can be written as:

$$\hat{x} = \arg\min_x \|\mathbf{A}x - y\|_2^2 + \lambda R(x), \tag{2}$$

where $\hat{x}$ is the estimated 3D volume, $\|\mathbf{A}x - y\|_2^2$ is the fidelity term that measures the discrepancy between the estimated measurements and the acquired measurements, $R(x)$ is the regularization term, which incorporates prior knowledge or assumptions about the image structure, such as smoothness, sparsity, or low-rank characteristics, depending on the specific imaging modality and application. Total variation (TV) (Rudin et al., 1992; Sidky & Pan, 2008) regularization is a common choice for the regularization term in medical image reconstruction, as it preserves the edges and structures of the image while reducing noise. The hyperparameter $\lambda$ balances the fidelity and regularization terms.

**Related Work** *(a) Sparse-View CT*. Classical CT reconstruction methods, such as Filtered Back Projection (FBP) and Iterative Reconstruction (IR), are incapable of handling the Sparse-View CT reconstruction problem. Modern deep learning methods have evolved from convolutional neural networks (CNNs) (Kang et al., 2017; Chen et al., 2017) to generative adversarial networks (GANs) (Yang et al., 2018) and, more recently, to diffusion-based models (Chung et al., 2022; 2023; Xu et al., 2024). Apart from optimization methods like NeRP (Shen et al., 2022), these models typically require large amounts of training data to achieve good performance. *(b) Compressed-Sensing MRI*. Traditional MRI reconstruction methods rely heavily on the Fourier Transform. However, the performance of Fourier Transform-based reconstruction decreases when the number of sampling points is reduced in Compressed-Sensing MRI. Similar to Sparse-View CT, deep learning methods in this field have evolved from CNNs (Zhu et al., 2018; Hyun et al., 2018) to GANs (Yang et al., 2017; Quan et al., 2018), and finally to diffusion-based models (Chung & Ye, 2022; Chung et al., 2023). A large amount of training data is also required to train these models. *(c) Relationship with Existing Works*. We categorize existing medical image reconstruction methods and compare their characteristics in Table 7. Recently, several contemporary works have adapted 3D Gaussian Splatting (3DGS) for CT reconstruction or novel view synthesis (Fu et al., 2024; Lin et al., 2024; Cai et al., 2025; Zha et al., 2024). 3DGR-CAR (Fu et al., 2024) incorporates U-Net (Ronneberger et al., 2015) to predefine Gaussian centers, which are then refined using 3DGS for the final reconstruction process. DIF-Gaussian (Lin et al., 2024) leverages 3D Gaussians to represent feature distributions, facilitating the estimation of attenuation coefficients. X-Gaussian redesigns a radiative Gaussian point cloud model for generating novel views in X-ray imaging applications. $R^2$-Gaussian (Zha et al., 2024) identifies shortcomings in the use of 3DGS for volumetric reconstruction and introduces an innovative approach to enhance volumetric reconstruction quality.

We emphasize the differences between our proposed method and these approaches. Unlike the above works, our GBIR is not based on 3DGS but introduces a novel Gaussian-based iterative method specifically tailored for medical image reconstruction. The entire process is end-to-end trainable and optimized for medical image reconstruction without any splatting or rendering processes.

## 3 GAUSSIAN-BASED ITERATIVE RECONSTRUCTION (GBIR)

Figure 2 illustrates our proposed GBIR method, which consists of two parts: **representation** and **reconstruction**. In the following sections, we provide detailed descriptions.

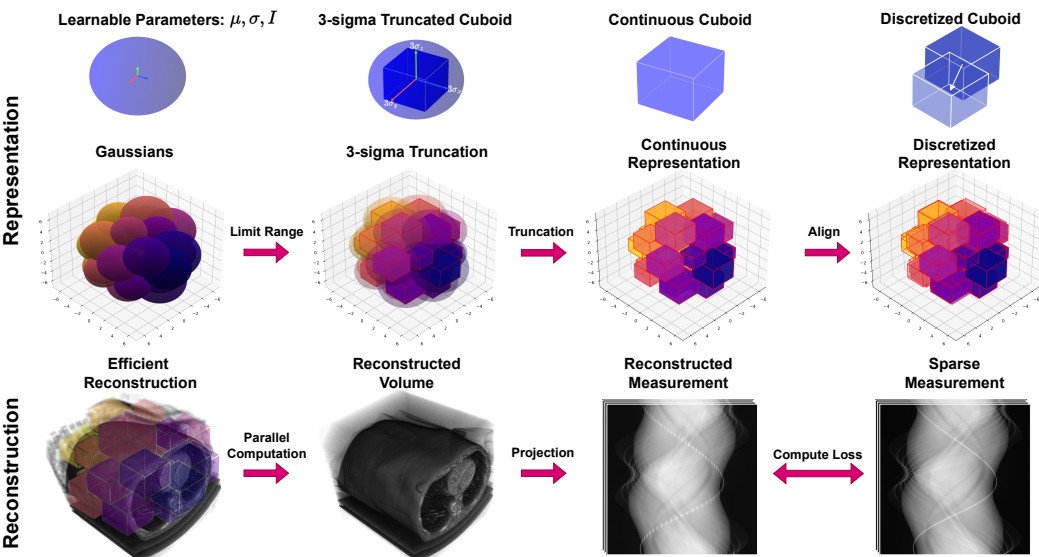

Figure 2: Our GBIR framework for medical image reconstruction. The 3D volume is represented by a set of 3D Gaussians, and the reconstruction process is conducted in an end-to-end manner.

## 3.1 TRUNCATED THREE-SIGMA GAUSSIAN REPRESENTATION

**Basic Formula.** We represent the 3D medical volume as the sum of a set of isotropic Gaussians. Each Gaussian function is characterized by its center at a mean value $\boldsymbol{\mu}$ and a covariance $\boldsymbol{\Sigma}$ where $\boldsymbol{\Sigma}$ is a diagonal matrix. We define the Gaussian function as follows:

$$G(\mathbf{x}, \boldsymbol{\mu}, \boldsymbol{\Sigma}) = \exp\left(-\frac{1}{2}(\mathbf{x} - \boldsymbol{\mu})^\top \boldsymbol{\Sigma}^{-1}(\mathbf{x} - \boldsymbol{\mu})\right), \tag{3}$$

where $\mathbf{x} \in \mathbb{R}^d$ represents a 3D point in the scene, exhibiting a bell-shaped curve symmetrically distributed around the mean $\boldsymbol{\mu}$. The spread of this function in the 3D space is determined by the standard deviation $\boldsymbol{\sigma}$.

Naively, we can formulate the reconstruction process of $n$ Gaussians as follows:

$$\mathbf{V} = \sum_{i=1}^{n} G(\mathbf{x}, \boldsymbol{\mu}_i, \boldsymbol{\Sigma}_i) \cdot I_i = \sum_{i=1}^{n} e^{-\frac{1}{2}(\mathbf{x} - \boldsymbol{\mu}_i)^\top \boldsymbol{\Sigma}_i^{-1}(\mathbf{x} - \boldsymbol{\mu}_i)} \cdot I_i = \sum_{i=1}^{n} e^{-\frac{1}{2}D_i^2} \cdot I_i, \tag{4}$$

In this equation, $I_i$ denotes the intensity of the $i$-th Gaussian. This intensity serves dual purposes: it represents the intensity of the voxel in the volume and also acts as the weight of the Gaussian. The term $(\mathbf{x} - \boldsymbol{\mu})^\top \boldsymbol{\Sigma}^{-1}(\mathbf{x} - \boldsymbol{\mu})$ is recognized as the squared Mahalanobis distance, and we denote it as $D_i^2$ for the $i$-th Gaussian for brevity.

However, this formulation is computationally expensive, as it requires the computation of the squared Mahalanobis distance for each voxel in the volume. To address this issue, we introduce a localized Gaussian mapping technique to accelerate the reconstruction process.

**Truncated Three-Sigma Gaussian** According to the Three-Sigma rule, in Gaussian distribution, the probability of a point falling within three standard deviations of the mean is approximately 99.73% (Appendix B). This implies that the influence of a Gaussian on a voxel diminishes as the distance from the Gaussian center to the voxel increases. By considering only the contributions of Gaussians within a specified proximity of each voxel, we can accelerate the reconstruction process.

Specifically, for each voxel in the 3D volume, we consider a neighborhood $\boldsymbol{\delta}$ around the voxel and compute the contributions of all Gaussians within this neighborhood. The contributions of all Gaussians within their neighborhoods are then added to their corresponding voxels in the volume. This process is repeated for all voxels in the volume, resulting in the final reconstructed 3D volume. The neighborhood around each voxel is centered at the Gaussian center.

Denote the target discretized 3D volume as $\mathbf{V} \in \mathbb{R}^{C \times H \times W}$ where $C$, $H$, and $W$ represent the size of the three dimensions, and denote the neighborhood around $i$-th Gaussian as $\boldsymbol{\delta}_i \in \mathbb{R}^{c \times h \times w \times d}$ where $c$, $h$, and $w$ represent the size of the neighborhood, $d = 3$ represents the dimension of 3D coordinates. Note the neighborhood is centered at the Gaussian center $\boldsymbol{\mu}_i$, thus the distance from the points in $\boldsymbol{\delta}_i$ to the center $\boldsymbol{\mu}_i$ is **a constant tensor** for all Gaussians[1], denoted as $\boldsymbol{\delta}' = \boldsymbol{\delta}_i - \boldsymbol{\mu}_i$ with broadcasting applied, where each point $\boldsymbol{p}$ in $\boldsymbol{\delta}_i$ and its corresponding point after transformation $\boldsymbol{p}'$ in $\boldsymbol{\delta}'_i$ satisfies $\boldsymbol{p}' = \boldsymbol{p} - \boldsymbol{\mu}_i$.

Hereby the computation of the squared Mahalanobis distance $D_i^2$ between the voxel and the Gaussian's mean can be simplified as:

$$D_i^2 = \boldsymbol{\delta}'^{\top} \boldsymbol{\Sigma}_i^{-1} \boldsymbol{\delta}'. \tag{5}$$

**Alignment and Differentiability** The computation above does not take the discretized grid into account, which is essential for the reconstruction process. The discretized 3D volume $\mathbf{V}$ is composed of integer coordinates, whereas $\boldsymbol{\mu}_i$ is continuous. Direct discretization of $\boldsymbol{\mu}_i$ to the nearest integer for indexing would render the reconstruction process non-differentiable. To address this, we compute each Gaussian's contribution at the discretized grid instead of its continuous position. We denote the $\boldsymbol{\delta}''_i$ as the discretized neighborhood around the Gaussian center. The relationship between $\boldsymbol{\delta}'_i$, $\boldsymbol{\delta}''_i$, and $\boldsymbol{\mu}_i$ is given by:

$$\boldsymbol{\delta}''_i = \boldsymbol{\delta}'_i - (\boldsymbol{\mu}_i - \lfloor \boldsymbol{\mu}_i \rfloor) = \boldsymbol{\delta}'_i - \Delta\boldsymbol{\mu}_i, \tag{6}$$

where we denote $\Delta\boldsymbol{\mu}_i = \boldsymbol{\mu}_i - \lfloor \boldsymbol{\mu}_i \rfloor$ for brevity. Each point $\boldsymbol{p}$ in $\boldsymbol{\delta}'_i$ and its corresponding point after transformation $\boldsymbol{p}'$ in $\boldsymbol{\delta}''_i$ satisfies $\boldsymbol{p}' = \boldsymbol{p} - (\boldsymbol{\mu}_i - \lfloor \boldsymbol{\mu}_i \rfloor)$. From now on, we use subscripts to denote the tensor dimensions to represent the broadcasting operations and tensor-wised operations. For example, Equation 6 will be written as $\boldsymbol{\delta}''_{n,c,h,w,d} = \boldsymbol{\delta}'_{c,h,w,d} - \Delta\boldsymbol{\mu}_{n,1,1,1,d}$. Here, $\boldsymbol{\delta}''_{n,c,h,w,d}$ is the tensor comprised of neighborhoods of all $n$ Gaussians, and $\Delta\boldsymbol{\mu}_{n,1,1,1,d}$ implicitly denotes the expansion of $\Delta\boldsymbol{\mu}_{n,d}$ to identical dimensions for element-wise subtraction.

## 3.2 Efficient Reconstruction

On the discretized 3D grid, the computation of the squared Mahalanobis distance tensor $D_{n,c,h,w}^2$ can be formulated as the Einstein summation:

$$D_{n,c,h,w}^2 = \sum_d \boldsymbol{\delta}''^{\top}_{n,c,h,w,d} \boldsymbol{\Sigma}^{-1}_{n,d,d} \boldsymbol{\delta}''_{n,c,h,w,d}. \tag{7}$$

By combining Equations 6 and 7, we decompose the large Einstein summation above into the sum of four smaller Einstein summations:

$$D_{n,c,h,w}^2 = \sum_d \boldsymbol{\delta}'_{c,h,w,d} \boldsymbol{\Sigma}^{-1}_{n,d,d} \boldsymbol{\delta}'_{c,h,w,d} - \sum_d \boldsymbol{\delta}'_{c,h,w,d} \boldsymbol{\Sigma}^{-1}_{n,d,d} \Delta\boldsymbol{\mu}_{n,1,1,d}$$
$$- \sum_d \Delta\boldsymbol{\mu}_{n,1,1,d} \boldsymbol{\Sigma}^{-1}_{n,d,d} \boldsymbol{\delta}'_{c,h,w,d} + \sum_d \Delta\boldsymbol{\mu}_{n,1,1,d} \boldsymbol{\Sigma}^{-1}_{n,d,d} \Delta\boldsymbol{\mu}_{n,1,1,d}. \tag{8}$$

Then we can compute the contributions of all Gaussians, denoted as $\Gamma_{n,c,h,w}$, as the following:

$$\Gamma_{n,c,h,w} = e^{-\frac{1}{2} D_{n,c,h,w}^2} \cdot I_n. \tag{9}$$

Note that $\Gamma_{n,c,h,w}$ is the contributions of all Gaussians within their neighborhoods, and the final step is to add up all the contributions to their corresponding voxels in the volume. A direct way is to loop over each Gaussian and add its contribution to the volume as $\mathbf{V}[\boldsymbol{\delta}_i] \leftarrow \mathbf{V}[\boldsymbol{\delta}_i] + \Gamma_i$. For acceleration, we use the parallel accumulation operation to compute the contributions of all Gaussians within their neighborhoods in parallel.

$$\mathbf{V}_{c,h,w} = \text{parallel\_accumulate}(\Gamma_{n,c,h,w}, \boldsymbol{\delta}_{n,c,h,w,d}). \tag{10}$$

---

[1]The shape of the Gaussian function remains invariant under translation; shifting the parameter $\mu$ changes the peak's location but does not alter the overall shape of the function.

## 3.3 OPTIMIZATION IN MEASUREMENT DOMAIN

After the 3D volume is reconstructed, we transform the 3D volume to the measurement domain and directly optimize it under the supervision of the current patient's measurement. The transformation $\mathcal{F}$ from the 3D volume to the measurement domain is achieved through the Radon transform for CT and Fourier transform for MRI[2].

$$\hat{\mathbf{M}} = \mathcal{F}(\mathbf{V}) = \begin{cases} Radon(\mathbf{V}), & \text{for CT} \\ Fourier(\mathbf{V}), & \text{for MRI} \end{cases} \tag{11}$$

where $\hat{\mathbf{M}}$ is the estimated measurement. Then the optimization problem becomes to minimize the discrepancy between the estimated measurement $\hat{\mathbf{M}}$ and the sparse measurement $\mathbf{M}$. We penalize the discrepancy in the measurement domain by $L_1$ norm and Structure Similarity Index (SSIM). Besides, we add a total variation (TV) regularization term to the 3D volume to preserve the structure of the volume. The optimization problem can be formulated as:

$$\min_{\mathbf{V}} \lambda_1 \left\| \hat{\mathbf{M}} - \mathbf{M} \right\|_1 + \lambda_2 (1 - \text{SSIM}(\hat{\mathbf{M}}, \mathbf{M})) + \lambda_3 \text{TV}(\mathbf{V}), \tag{12}$$

where $\lambda_1 = 1$, $\lambda_2 = 500$, and $\lambda_3 = 500$ are hyperparameters to balance the three terms. For the MRI reconstruction, the measurement $M$ and the estimated measurement $\hat{M}$ are complex-valued, and we compute the loss separately for the real and imaginary parts and sum them up. Iteratively, we update the parameters of the Gaussians to minimize the objective function. The optimization process is conducted in an end-to-end manner, and the final 3D volume is obtained after convergence.

## 4 MULTI-ORGAN MEDICAL IMAGE RECONSTRUCTION DATASET (MORE)

Existing datasets usually focus on a single body part or disease, which substantially hinders a more thorough and comprehensive assessment of current research on medical image reconstruction. Advanced methods including (Chung et al., 2023; Xu et al., 2024; Yang et al., 2022; Xia et al., 2022) usually evaluate on a single body part, such as only abdomen part in AAPM-Mayo LDCT Challenge Dataset (Moen et al., 2021), or only brain part in BRATS (Menze et al., 2014) dataset. It is difficult to conclude the effectiveness of a method based solely on the results of a single body part, and its generalization ability remains to be verified. To address this limitation, we propose the **M**ulti-**O**rgan Medical Image **RE**construction (**MORE**) dataset, which has the following characteristics:

- It incorporates both CT and MRI data types, and a diverse set of body parts. To be specific, MORE contains over 65,755 CT slices and 7,498 MRI slices from 173 patients, covering 15 body parts in CT scans and 5 body parts in MRI scans. Table 1 presents a detailed comparison of MORE with existing medical image reconstruction datasets.
- MORE exhibits a rational distribution of demographics and diseases. To be specific, MORE involves a total of 173 patients and 189 examinations. Some patients underwent multiple examinations, resulting in 135 CT scans and 54 MRI scans. The median age of the participants was 52 years, ranging from 7 to 85 years. The age distribution is as follows: 0-20 years (5.4%), 21-40 years (29.5%), 41-60 years (37.2%), 61-80 years (24.0%), 81-100 years (3.9%). The gender distribution was 59.7% male and 40.3% female. MORE contains 25 types of diseases in CT and 17 types of diseases in MRI, respectively. We show the specific distribution of the CT and MRI scans in Figure 3 and Figure 4, and provide some samples in Figure 6 for visualization.
- MORE has been approved by the ethics committee of corresponding hospital, and the approval number also has been obtained[3]. All DICOM data has been anonymized by RSNA clinical trial processor to protect patient privacy and comply with the Helsinki declaration. We will release the dataset for public availability.

  Currently, MORE dataset provides DICOM images and does not include the original raw measurements. This aligns with common practices in medical image reconstruction research as demonstrated by several advanced methods (Yang et al., 2022; Chung et al., 2023; Xu et al., 2024), which often rely on simulated measurements generated from image slices. In the experiments, we simulate measurements by applying the Radon transform for CT data and the Fourier transform for MRI data following previous research (Xu et al., 2024; Chung et al., 2023).

---

[2]In this paper, we only reconstruct the magnitude of the MRI image, which is real-valued.

[3]Due to the double-blind policy, the information of the hospital will be disclosed after the review process.

Table 1: Comparison of MORE with existing medical image reconstruction datasets. 'New Source' means whether the dataset is collected from new source or not.

| Dataset | #Body Parts | #Images | New Source |
|---|---|---|---|
| MORE | **15** (CT) / **5** (MRI) | 65,755 CT, 7,498 MRI | ✓ |
| AAPM-Mayo LDCT (Moen et al., 2021) | 3 (chest, abdomen, head) | 25,141 CT | ✓ |
| LoDoPaB-CT (Leuschner et al., 2021) | 1 (chest) | 46,573 CT | × |
| Covidx-CT (Gunraj et al., 2020) | 1 (chest) | 104,009 CT | × |
| LIDC/IDRI (Armato III et al., 2011) | 1 (chest) | 1,018 CT | ✓ |
| FUMPE (Masoudi et al., 2018) | 1 (chest) | 8,792 CT | ✓ |
| JSRT (Shiraishi et al., 2000) | 1 (chest) | 247 CT | ✓ |
| Fast MRI (Knoll et al., 2020) | 3 (knee, brain, prostate) | 167,375 MRI | ✓ |
| SKM-TEA (Desai et al., 2022) | 1 (knee) | 25,000 MRI | ✓ |
| Calgary-Campinas-359 (Souza et al., 2018) | 1 (brain) | 42,752 MRI | ✓ |
| BraTS (Menze et al., 2014) | 1 (brain) | 57,195 MRI | ✓ |

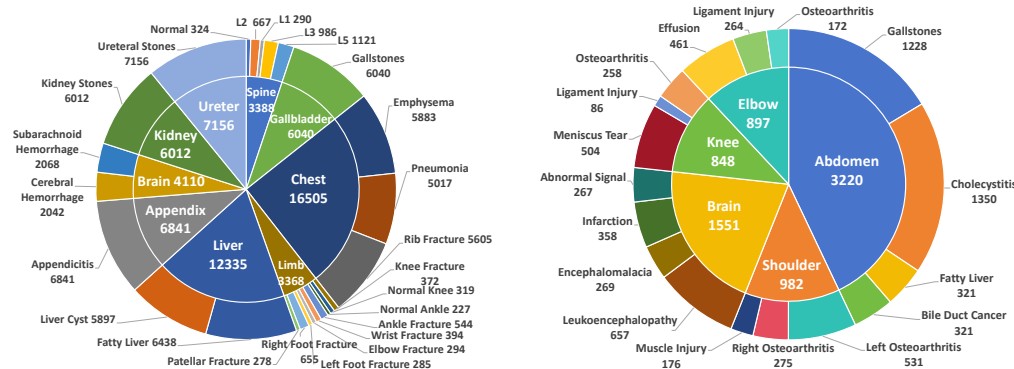

Figure 3: Data distribution of MORE CT part, containing 15 organs and 25 disease types.

Figure 4: Data distribution of MORE MRI part, containing 5 organs and 17 disease types.

## 5 EXPERIMENTS

In this section, we extensively evaluate and benchmark various types of methods on both public widely used datasets and our newly proposed MORE dataset.

### 5.1 EXPERIMENTAL SETTINGS

**Datasets** We extensively benchmark various methods on both public widely used datasets and our newly proposed MORE dataset. For the public dataset, we use the widely-used AAPM-Mayo LDCT Challenge Dataset (Moen et al., 2021) for CT reconstruction, and the BRATS dataset (Menze et al., 2014) for MRI reconstruction with those learning-based methods pretrained on the fastMRI dataset (Knoll et al., 2020) following the setting of our baseline method DiffusionMBIR (Chung et al., 2023). We simulate the sparse-view CT reconstruction of fan-beam geometry with 60, 90, 120, and 180 views, and the MRI reconstruction by 1D uniform distribution with an acceleration factor of 2 to subsample the $k$-space data.

**Evaluation Metrics** We follow the standard practice in medical image reconstruction (Chung et al., 2023; Xu et al., 2024; Chen et al., 2017) to evaluate the performance of different methods using the Peak Signal-to-Noise Ratio (PSNR) (Hore & Ziou, 2010) and the Structural Similarity Index (SSIM) (Wang et al., 2004). The detailed definitions of PSNR and SSIM are provided in Appendix A.

**Compared Methods** We compare our proposde GBIR method with different types of baselines which covering representative methods to state-of-the-art methods. We choose traditional methods FBP (Bracewell & Riddle, 1967) and IFFT (Gallagher et al., 2008) that widely used in clinical practice, early DLR methods REDCNN (Chen et al., 2017), AUTOMAP (Zhu et al., 2018), and the 3D Scene-based method NeRP (Shen et al., 2022) that implicitly learn the prior from the data, and

Table 2: Efficiency of different methods in terms of time and GPU memory consumption during training and inference.'Train' and 'Inference' are denoted as 'T.' and 'Inf.', respectively. MCG and DiffusionMBIR share the same score function and thus have the same training time and memory consumption.

| Method | T. Time (min) | T. Mem (MiB) | Inf. Time (min) | Inf. Mem (MiB) |
|---|---|---|---|---|
| RED-CNN (Chen et al., 2017) | 221 | 4857 | 2.4 | 1665 |
| AUTOMAP (Zhu et al., 2018) | 33.4 | 9.75 | ≈ 0.2 | 49140 |
| Score-MRI (Chung & Ye, 2022) | 9833 | 6143 | 1941 | 16685 |
| MCG (Chung et al., 2022) | 10342 | 7103 | 3290 | 7392 |
| DiffusionMBIR (Chung et al., 2023) | 10342 | 7103 | 1983 | 16673 |
| SWORD (Xu et al., 2024) | 3017 | 16580 | 5094 | 3051 |
| NeRP (Shen et al., 2022) | 0 | 0 | 1121 | 44927 |
| GBIR (Ours) | 0 | 0 | 464 | 34126 |

Table 3: SV-CT reconstruction on AAPM-Mayo LDCT dataset. Best in **Bold.**

| Method | Extra Data | 180-view | | 120-view | | 90-view | | 60-view | |
|---|---|---|---|---|---|---|---|---|---|
| | | PSNR | SSIM | PSNR | SSIM | PSNR | SSIM | PSNR | SSIM |
| FBPConvNet (Jin et al., 2017) | ✓ | 42.23 | 0.988 | 39.45 | 0.983 | 37.11 | 0.976 | 35.63 | 0.966 |
| U-Net (TRPMS 18) (Lee et al., 2018) | ✓ | 38.37 | 0.985 | 35.58 | 0.977 | 30.09 | 0.947 | 28.83 | 0.937 |
| PLANet (ACM'MM 22) (Yang et al., 2022) | ✓ | 42.76 | 0.965 | 41.67 | 0.962 | 40.99 | 0.957 | 38.97 | 0.941 |
| DDPM (Xia et al., 2022) | ✓ | 40.95 | 0.985 | 37.90 | 0.976 | 35.15 | 0.963 | 32.04 | 0.934 |
| MCG (Chung et al., 2022) | ✓ | 40.42 | 0.969 | 39.57 | 0.960 | 38.02 | 0.935 | 37.17 | 0.921 |
| DiffusionMBIR (Chung et al., 2023) | ✓ | 41.78 | 0.990 | 40.83 | 0.964 | 39.98 | 0.942 | 38.67 | 0.932 |
| GMSD (TRPMS 23) (Guan et al., 2023) | ✓ | 41.44 | 0.988 | 39.41 | 0.981 | 37.25 | 0.974 | 34.31 | 0.958 |
| SWORD (Xu et al., 2024) | ✓ | 45.08 | 0.994 | 42.49 | 0.990 | 41.27 | 0.986 | 38.49 | 0.978 |
| FBP (Bracewell & Riddle, 1967) | ✗ | 31.69 | 0.882 | 28.30 | 0.787 | 26.20 | 0.701 | 23.18 | 0.595 |
| GBIR (Ours) | ✗ | **46.39** | **0.995** | **45.24** | **0.994** | **43.21** | **0.991** | **40.17** | **0.985** |

most advanced diffusion-based DLR methods DiffusionMBIR (Chung et al., 2023), MCG (Chung et al., 2022), score-mri (Chung & Ye, 2022), SWORD (Xu et al., 2024).

**Experimental Settings** As shown in Table 7, most methods, particularly DLR methods, require the entire training dataset to learn parameters. We mark these methods as requiring 'Extra Data'. For other optimization-based methods, including FBP (Bracewell & Riddle, 1967), IFFT (Gallagher et al., 2008), NeRP (Shen et al., 2022), and our GBIR, we directly evaluate the performance on the test set without using any training data. All experiments are conducted on an Ubuntu server equipped with an NVIDIA RTX 6000 Ada Generation GPU with 48 GiB of memory.

**Hyperparameter Setting** For our proposed GBIR framework, we initialize the number of Gaussians to 150. We use the Adam optimizer with a learning rate of 3e-4 and decay to 3e-5 at the end of training. For 60-view, 90-view, 120-view, and 180-view SV-CT, we set the training iteration to 5K, 6K, and 7K, and 10K, respectively. For the CS-MRI, we set the training iteration to 3K.

## 5.2 Sparse-View CT on AAPM-Mayo LDCT Dataset

AAPM-Mayo LDCT Challenge Dataset is widely used for Sparse-View CT reconstruction, and we follow the latest state-of-the-art method SWORD (Xu et al., 2024) to conduct the evaluation with 60-view, 90-view, 120-view, and 180-view, which is also a common setting adopted (Guan et al., 2023; Yang et al., 2022). The results are shown in Table 3. Our proposed GBIR outperforms all the compared methods in terms of PSNR and SSIM without bells and whistles.

## 5.3 Compressed-Sensing MRI on BRATS Dataset

We evaluate the performance of different methods on the BRATS dataset for CS-MRI reconstruction following the setting in Chung et al. (2023); Chung & Ye (2022). The result of DuDoRNet (Lahiri

Table 4: CS-MRI reconstruction on BRATS dataset. Best in **Bold.**

| Method | Extra Data | Axial | | Coronal | | Sagittal | |
|---|---|---|---|---|---|---|---|
| | | PSNR | SSIM | PSNR | SSIM | PSNR | SSIM |
| RED-CNN (Chen et al., 2017) | ✓ | 33.23 | 0.920 | 29.11 | 0.916 | 28.91 | 0.910 |
| Unet (Lee et al., 2018) | ✓ | 37.15 | 0.929 | 31.56 | 0.899 | 30.90 | 0.816 |
| DuDoRNet (Lahiri et al., 2023) | ✓ | 39.78 | 0.974 | 33.56 | 0.927 | 33.48 | 0.927 |
| AUTOMAP (Zhu et al., 2018) | ✓ | 31.11 | 0.913 | 30.96 | 0.905 | 29.39 | 0.895 |
| ScoreMRI (Chung & Ye, 2022) | ✓ | 40.38 | 0.968 | 33.97 | 0.925 | 34.02 | 0.928 |
| DiffusionMBIR (Chung et al., 2023) | ✓ | **41.49** | **0.974** | 37.36 | 0.942 | 37.18 | 0.953 |
| IFFT (Gallagher et al., 2008) | × | 32.15 | 0.914 | 31.80 | 0.911 | 31.44 | 0.910 |
| GBIR (Ours) | × | 40.40 | 0.973 | **39.64** | **0.969** | **39.45** | **0.968** |

Table 5: Effectiveness of Three-Sigma Gaussian Truncation.

| | PSNR | SSIM | Inference Time (h) | Space Consumption (GiB) |
|---|---|---|---|---|
| $11 \times 11 \times 11$ | 44.64 | 0.989 | 5.88 | 32.24 |
| $13 \times 13 \times 13$ | 45.38 | 0.991 | 8.10 | 38.22 |
| $15 \times 15 \times 15$ | 46.05 | 0.993 | 11.46 | 47.53 |
| $Three - Sigma$ | **46.39** | **0.995** | 7.73 | 33.32 |

et al., 2023) is sourced from Chung et al. (2023). Different from CT, MRI scans are usually conducted in three different directions, including axial, coronal, and sagittal. The results are shown in Table 4. Our proposed GBIR achieves the best performance in coronal and sagittal views, and the second-best performance in the axial view, which is only slightly lower than the best method. Besides, GBIR shows a more balanced performance across different views compared to other methods.

### 5.4 BENCHMARK AND FINDINGS ON MORE DATASET

We benchmark the performance of different methods on the newly proposed MORE dataset, which contains a wide range of body parts and diseases. Our benchmark include 15 body parts and 25 diseases for CT scans (Table 9 to Table 23), and 5 body parts and 17 diseases for MRI scans (Table 24 to Table 28). To the best of our knowledge, this is the first time that such a comprehensive dataset is used for evaluating medical image reconstruction methods. More than comparing the performance, we also provide some insights and findings from the benchmark on the MORE dataset.

*Optimization-based methods are more robust to the influence of data.*: We observe that optimization-based methods, including FBP (Bracewell & Riddle, 1967), IFFT (Gallagher et al., 2008), NeRP (Shen et al., 2022), and our GBIR, show consistent performance across different body parts and diseases. In contrast, learning-based methods, such as REDCNN (Chen et al., 2017), AUTOMAP (Zhu et al., 2018), and score-mri (Chung & Ye, 2022), show a more significant performance variation across different body parts and diseases. This indicates that optimization-based methods are more robust to the influence of data, while learning-based methods are more sensitive to the data distribution.

*A comprehensive dataset helps improve the generalization ability for learning-based methods*: Table 8 and Table 23 both are evaluated on the subarachnoid hemorrhage disease, but their training data are different. The former is trained on the AAPM-Mayo LDCT dataset, while the latter is trained on the MORE dataset. We observe that the learning-based methods, including REDCNN (Chen et al., 2017), AUTOMAP (Zhu et al., 2018), and score-mri (Chung & Ye, 2022), show better performance when trained on the MORE dataset compared to the AAPM-Mayo LDCT dataset. This indicates that a comprehensive dataset with diverse body parts and diseases can help improve the generalization ability of learning-based methods.

*Significant performance variation across different body parts*: We observe that the performance of different methods varies significantly across different body parts and diseases. Figure 7 and Figure 8 show the distribution of PSNR and SSIM across different body parts and diseases for CT and MRI scans, respectively. We observe that the performance of different methods varies significantly across different body parts and diseases. For example, the performance of RED-CNN is considerably

Table 6: Effectiveness of Efficient Reconstruction.

|  | Direct Reconstruction | Non-Parallel Reconstruction | Efficient Reconstruction |
|---|---|---|---|
| Rendering Time (s) | 1.03-1.12 | 0.98-1.09 | 0.09-0.12 |
| Space Consumption (GiB) | 47.98 | 33.32 | 33.32 |

lower on the **Emphysema** part compared to the **Ureteral Calculi** part. It is important to evaluate the performance on a diverse dataset with multiple organs to ensure the robustness of the method.

Furthermore, our GBIR method shows the best performance across different body parts and diseases on the MORE dataset, which demonstrates the effectiveness and robustness of our proposed method.

### 5.5 ABLATION STUDY AND EFFICIENCY ANALYSIS

To avoid confusion with the training time, inference time, and rendering time, here the training time refers to the time consumed for training the neural network for DLR methods, while the inference time refers to the time consumed for reconstructing the volume. Thus, NeRP (Shen et al., 2022) and our GBIR do not have training time, as they do not require any training data. The rendering time refers to the time consumed for reconstructing the volume from the Gaussians.

**Computational Efficiency** We provide an efficiency analysis of different methods in terms of training time, GPU memory consumption, inference time, and GPU memory consumption. The results are shown in Table 2. Our proposed GBIR achieves the best efficiency in terms of training time and GPU memory consumption, as it does not require any training data. In contrast, learning-based methods, such as RED-CNN (Chen et al., 2017), score-mri (Chung & Ye, 2022), and MCG (Chung et al., 2022), require a large amount of training data and thus consume more training time and GPU memory. For inference time and GPU memory consumption, our proposed GBIR achieves better efficiency than the advanced diffusion-based methods and NeRP, which is essential for real-time applications in clinical practice.

**Effect of Three-Sigma Truncation** We evaluate the effect of Three-Sigma truncation in the GBIR framework on AAPM-Mayo LDCT dataset 180-view SV-CT. We substitute the Three-Sigma truncation with different size of cuboid box, including $11 \times 11 \times 11$, $13 \times 13 \times 13$, and $15 \times 15 \times 15$. The results are shown in Table 5.

**Effect of Efficient Reconstruction** In Table 6, we compare the rendering time and space consumption of direct reconstruction, non-parallel reconstruction, and efficient reconstruction. Specifically, direct reconstruction refers to the reconstruction with formula 7, non-parallel reconstruction refers to the reconstruction process without parallel computation.

## 6 DISCUSSION

**Limitations** It should be noted that although our proposed GBIR is faster than the advanced methods, it is still slower than the traditional DLR methods. In Table 2, the inference time of RED-CNN and AUTOMAP is much shorter than all other advanced methods. Nevertheless, their performance is significantly lower. This trade-off between speed and performance is a common challenge in medical image reconstruction, and it remains an open problem for future research.

**Visualization** We include a visualization of the reconstruction process in Appendix 5 to provide a better understanding of the reconstruction process of our proposed GBIR method. Iteration by iteration, the reconstruction becomes clearer and more detailed. We also provide histograms of PSNR on the MORE dataset of 60-view SV-CT and axial CS-MRI in Figure 7 and Figure 8, respectively. The histograms show the distribution of PSNR across different body parts and diseases, providing insights into the performance of different methods on the MORE dataset.

**Conclusion** In this paper, we present a novel Gaussian-based image reconstruction method, GBIR, which achieves state-of-the-art performance on both public widely used datasets and our newly proposed MORE dataset. Our proposed method is efficient and robust, making it suitable for tailored reconstruction of different body parts and diseases. We also provide a comprehensive benchmark on the MORE dataset, which includes a wide range of body parts and diseases, to facilitate further research in medical image reconstruction.

## ETHICS STATEMENT

We make the following ethical considerations in our work:

- Our proposed MORE dataset has been collected with the approval of the hospital ethics committee.
- All information that could potentially identify patients has been removed from the dataset to ensure patient privacy.
- All other datasets used in our work are publicly available and have been used in accordance with the terms of use.
- We have followed the standard practice in medical image reconstruction and have conducted our experiments in a responsible and ethical manner.
- We have provided a detailed description of our methods and results to ensure transparency and reproducibility.
- We will make our code and data publicly available to facilitate further research and ensure transparency.

## REPRODUCIBILITY STATEMENT

We provide the following information to facilitate the reproducibility of our work:

- We include the metadata of the MORE dataset in the supplementary material for reference.
- We have provided the detailed experimental results and evaluation metrics in the paper to ensure transparency and reproducibility.
- After the double-blind review process, we will make our code and data publicly available to facilitate further research and ensure transparency.

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

# Appendix

## A  EVALUATION METRICS

In this section, we describe the evaluation metrics used in the paper.

### A.1  PEAK SIGNAL-TO-NOISE RATIO (PSNR)

The PSNR (Hore & Ziou, 2010) is a widely used metric to evaluate the quality of the reconstructed images. It is defined as:

$$\text{PSNR}(x, y) = 10 \cdot \log_{10} \left( \frac{\text{MAX}^2}{\text{MSE}(x, y)} \right), \tag{13}$$

where MAX is the maximum possible pixel value of the image and $\text{MSE}(x, y)$ is the mean squared error between the original and reconstructed images.

### A.2  STRUCTURAL SIMILARITY INDEX (SSIM)

The SSIM (Wang et al., 2004) is a metric that measures the similarity between two images. It is defined as:

$$\text{SSIM}(x, y) = \frac{(2\mu_x\mu_y + C_1)(2\sigma_{xy} + C_2)}{(\mu_x^2 + \mu_y^2 + C_1)(\sigma_x^2 + \sigma_y^2 + C_2)}, \tag{14}$$

where $\mu_x$ and $\mu_y$ are the mean values of the images $x$ and $y$, $\sigma_x^2$ and $\sigma_y^2$ are the variances of the images, $\sigma_{xy}$ is the covariance of the images, and $C_1$ and $C_2$ are constants to stabilize the division with weak denominator.

## B  THREE-SIGMA RULE

The *Three-Sigma rule* states that approximately $99.73\%$ of the data in a Gaussian distribution lies within three standard deviations of the mean. This result is derived from the properties of the Gaussian (normal) distribution.

For a random variable $X$ that follows a Gaussian distribution with mean $\mu$ and standard deviation $\sigma$, the probability density function (PDF) is given by:

$$f(x) = \frac{1}{\sigma\sqrt{2\pi}} \exp \left( -\frac{(x - \mu)^2}{2\sigma^2} \right)$$

To find the probability that $X$ lies within three standard deviations of the mean, i.e., within the interval $[\mu - 3\sigma, \mu + 3\sigma]$, we compute the following probability:

$$P(\mu - 3\sigma \leq X \leq \mu + 3\sigma) = \int_{\mu-3\sigma}^{\mu+3\sigma} \frac{1}{\sigma\sqrt{2\pi}} \exp \left( -\frac{(x - \mu)^2}{2\sigma^2} \right) dx$$

To simplify the integral, we standardize the normal distribution by defining a standard normal variable $z$ as:

$$z = \frac{x - \mu}{\sigma}$$

This transforms the limits of the integral from $[\mu - 3\sigma, \mu + 3\sigma]$ to $[-3, 3]$. The PDF of the standard normal distribution is then:

$$f(z) = \frac{1}{\sqrt{2\pi}} \exp \left( -\frac{z^2}{2} \right)$$

Table 7: Different Types of Medical Image Reconstruction Methods.

| | Representative Methods | Full | Sparse | Trainable | Data Indep. | Inf. Speed |
|---|---|---|---|---|---|---|
| Traditional | FBP (Bracewell & Riddle, 1967) | ✓ | ✗ | ✗ | ✓ | Real-Time |
| | IFFT (Gallagher et al., 2008) | ✓ | ✓ | ✗ | ✓ | Real-Time |
| Direct Learning | AUTOMAP (Zhu et al., 2018) | ✓ | ✓ | ✓ | ✗ | Very Fast |
| | iRadonMAP (He et al., 2020) | ✓ | ✓ | ✓ | ✗ | Very Fast |
| Image-Based Denoising | FBPConvNet (Jin et al., 2017) | ✓ | ✓ | ✓ | ✗ | Very Fast |
| | REDCNN (Chen et al., 2017) | ✓ | ✓ | ✓ | ✗ | Very Fast |
| Dual-Domain Reconstruction | HDNet (Hu et al., 2020) | ✓ | ✓ | ✓ | ✗ | Very Fast |
| Diffusion-Based DLR | MCG (Chung et al., 2022) | ✓ | ✓ | ✓ | ✗ | Low |
| | DiffusionMBIR (Chung et al., 2023) | ✓ | ✓ | ✓ | ✗ | Low |
| | SWORD (Xu et al., 2024) | ✓ | ✓ | ✓ | ✗ | Low |
| 3D Scene Reconstruction | NeRP (Shen et al., 2022) | ✓ | ✓ | ✓ | ✓ | Medium |
| | DIFGaussian (Lin et al., 2024) | ✓ | ✓ | ✓ | ✓ | Fast |
| | 3DGR-CAR (Fu et al., 2024) | ✓ | ✓ | ✓ | ✓ | Fast |
| | X-Gaussian (Lin et al., 2024) | ✓ | ✓ | ✓ | ✓ | Fast |
| | $R^2$-Gaussian (Zha et al., 2024) | ✓ | ✓ | ✓ | ✓ | Fast |
| Our method | GBIR | ✓ | ✓ | ✓ | ✓ | Fast |

Thus, the probability becomes:

$$P(-3 \leq z \leq 3) = \int_{-3}^{3} \frac{1}{\sqrt{2\pi}} \exp\left(-\frac{z^2}{2}\right) dz$$

This integral does not have a closed-form solution but can be numerically approximated. Using standard numerical methods or precomputed values from the cumulative distribution function (CDF) of the standard normal distribution, the result of this integral is approximately:

$$P(-3 \leq z \leq 3) \approx 0.9973$$

This confirms that approximately 99.73% of the data in a Gaussian distribution lies within three standard deviations from the mean.

The contribution of a Gaussian distribution decreases rapidly as the distance from its mean increases. Therefore, in medical image reconstruction, truncating the Gaussian distribution at three standard deviations from the mean can remove the negligible tail values while retaining the majority of the distribution.

## C   VISUALIZATION

Figure 5 shows the gradual convergence of the GBIR framework for a brain CT reconstruction. The 3D volume is gradually reconstructed from the initial random noise to the final clear structure. The convergence process is conducted in an end-to-end manner, and the final 3D volume is obtained after convergence.

## D   DATA ACQUISITION AND PROCESSING

**Staff Configuration** All CT and MRI scans were collected and evaluated by three experienced radiologists. The radiologists were responsible for reviewing the scans and identifying any abnormalities or diseases. Among the three radiologists, two were senior radiologists with over 10 years of experience, and one was a junior radiologist with 5 years of experience. The radiologists worked together to ensure the accuracy and consistency of the data.

**Data Selection** The CT and MRI scans were selected based on the following criteria: (1) the scans were of high quality, with minimal artifacts or noise, (2) the scans covered a wide range of body parts and conditions, and (3) the scans were representative of the clinical cases encountered in practice.

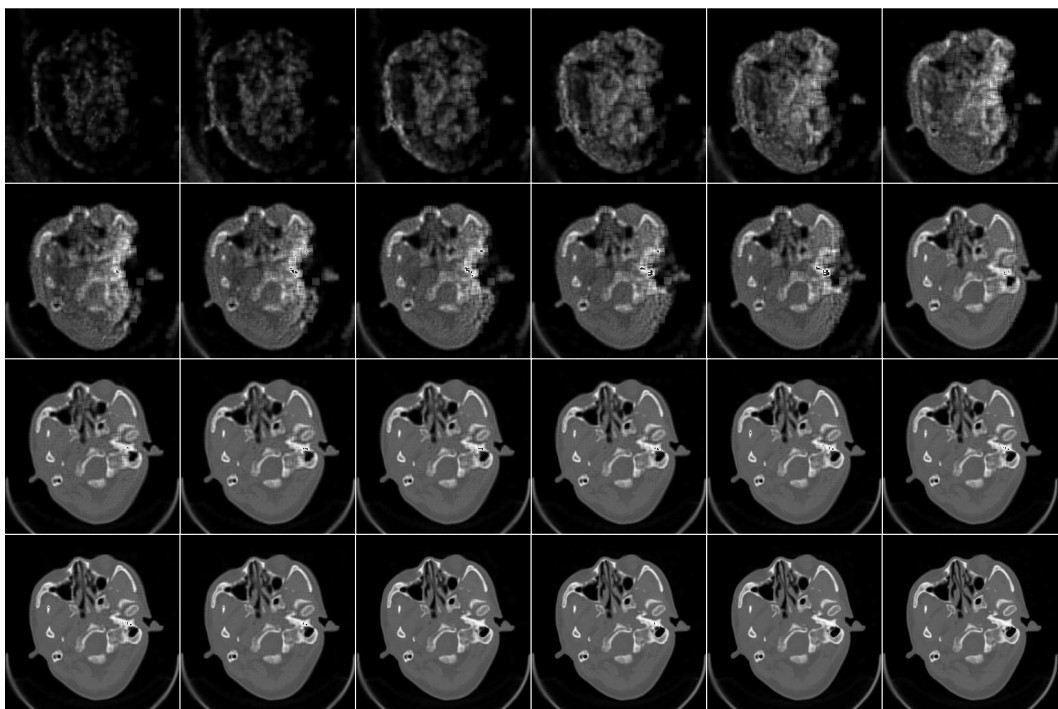

Figure 5: Iterative reconstruction visualization of our GBIR.

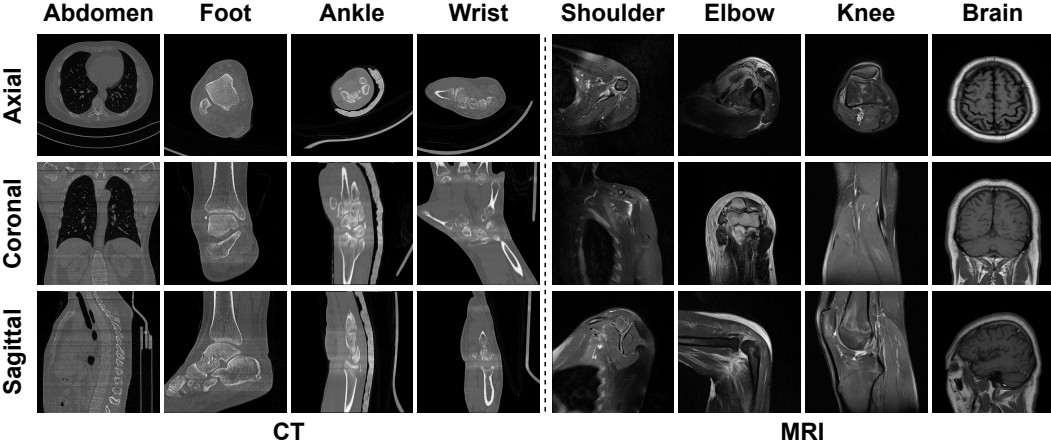

Figure 6: Examples of MORE dataset, containing CT and MRI scans from 4 different organs.

In practice, the radiologists first categorized the scans based on the body part imaged and the condition depicted, and then select typical cases from the corresponding parts, including internal and external medicine and acute and chronic cases.

**Scan Parameters** Each individual sample selects the window width and window position that are commonly displayed for the corresponding disease type. Samples of two slice thicknesses (1mm and 3mm) are chosen for CT scans, and two echo times (TE) are chosen for MRI scans. The MRI scans are collected using a 1.5T MRI scanner.

**Data processing** The image data is provided and easy to use. Slices within the same sequence can be identified with file names, and each slice is stored as a 2D array of pixel intensities without extra transformation. Intensity values depend on the type of scan (CT or MRI) and the scanning

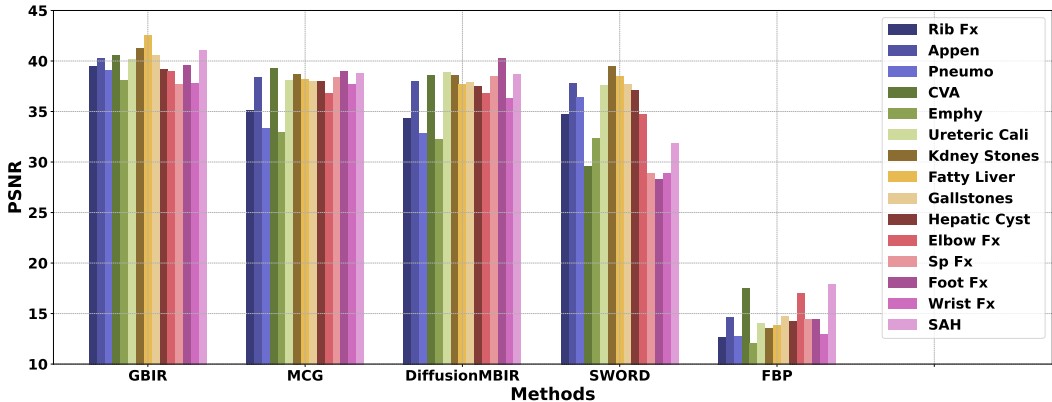

Figure 7: Performance of various methods on different organs within our MORE dataset, evaluated by the PSNR metric on 60 view SV-CT.

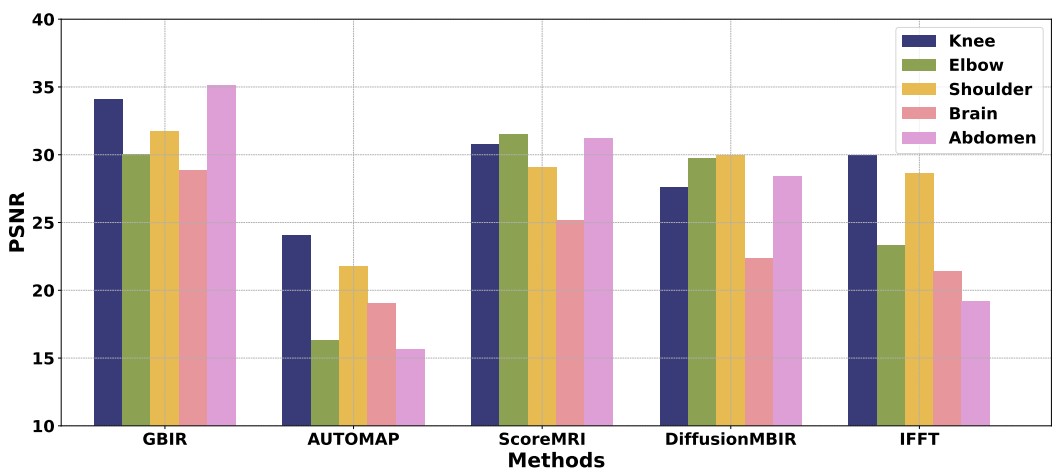

Figure 8: Performance of various methods on different organs within our MORE dataset, evaluated by the PSNR metric on axial CS-MRI.

parameters. For CT scans, the pixel intensities represent Hounsfield units, while for MRI scans, the pixel intensities represent signal intensities. To facilitate other researchers' use, we also provide PNG images for each DICOM file which can be easily visualized.

Table 8: SV-CT reconstruction of MORE dataset **Subarachnoid Hemorrhage** trained on AAPM-Mayo LDCT Dataset. Best in **Bold.**

| Method | Extra Data | 180-view | | 120-view | | 90-view | | 60-view | |
|---|---|---|---|---|---|---|---|---|---|
| | | PSNR | SSIM | PSNR | SSIM | PSNR | SSIM | PSNR | SSIM |
| RED-CNN (Chen et al., 2017) | ✓ | 28.03 | 0.818 | 27.76 | 0.795 | 27.43 | 0.792 | 26.40 | 0.787 |
| MCG (Chung et al., 2022) | ✓ | 35.85 | 0.874 | 35.90 | 0.875 | 35.78 | 0.870 | 35.59 | 0.869 |
| DiffusionMBIR (Chung et al., 2023) | ✓ | 36.65 | 0.962 | 36.59 | 0.962 | 36.57 | 0.963 | 36.02 | 0.961 |
| SWORD (Xu et al., 2024) | ✓ | 38.03 | 0.971 | 37.36 | 0.954 | 32.42 | 0.885 | 28.83 | 0.813 |
| FBP (Bracewell & Riddle, 1967) | ✗ | 21.31 | 0.440 | 20.84 | 0.423 | 19.22 | 0.404 | 17.93 | 0.361 |
| NeRP (Shen et al., 2022) | ✗ | 23.72 | 0.760 | 23.34 | 0.760 | 23.84 | 0.800 | 24.04 | 0.791 |
| GBIR (Ours) | ✗ | **43.29** | **0.993** | **42.74** | **0.993** | **41.98** | **0.992** | **41.02** | **0.992** |

Table 9: SV-CT reconstruction on **Emphysema**. Best in **Bold.**

| Method | Extra Data | 180-view | | 120-view | | 90-view | | 60-view | |
|---|---|---|---|---|---|---|---|---|---|
| | | PSNR | SSIM | PSNR | SSIM | PSNR | SSIM | PSNR | SSIM |
| RED-CNN (Chen et al., 2017) | ✓ | 29.58 | 0.714 | 28.44 | 0.644 | 27.06 | 0.623 | 27.27 | 0.588 |
| MCG (Chung et al., 2022) | ✓ | 32.72 | 0.820 | 32.84 | 0.821 | 34.47 | 0.843 | 32.90 | 0.820 |
| DiffusionMBIR (Chung et al., 2023) | ✓ | 32.58 | 0.933 | 32.64 | 0.936 | 32.45 | 0.932 | 32.24 | 0.932 |
| SWORD (Xu et al., 2024) | ✓ | 35.38 | 0.879 | 34.52 | 0.864 | 33.78 | 0.849 | 32.30 | 0.827 |
| FBP (Bracewell & Riddle, 1967) | ✗ | 18.55 | 0.365 | 16.29 | 0.293 | 14.77 | 0.248 | 12.03 | 0.193 |
| NeRP (Shen et al., 2022) | ✗ | 25.41 | 0.744 | 25.21 | 0.735 | 25.40 | 0.745 | 25.39 | 0.745 |
| GBIR (Ours) | ✗ | **39.47** | **0.950** | **39.04** | **0.946** | **38.42** | **0.941** | **38.04** | **0.937** |

Table 10: SV-CT reconstruction on **Ureteral Calculi**. Best in **Bold.**

| Method | Extra Data | 180-view | | 120-view | | 90-view | | 60-view | |
|---|---|---|---|---|---|---|---|---|---|
| | | PSNR | SSIM | PSNR | SSIM | PSNR | SSIM | PSNR | SSIM |
| RED-CNN (Chen et al., 2017) | ✓ | 37.04 | 0.901 | 35.63 | 0.913 | 32.07 | 0.759 | 31.46 | 0.844 |
| MCG (Chung et al., 2022) | ✓ | 37.94 | 0.901 | 37.99 | 0.901 | 38.04 | 0.902 | 38.05 | 0.902 |
| DiffusionMBIR (Chung et al., 2023) | ✓ | 38.37 | 0.968 | 38.24 | 0.967 | 38.13 | 0.967 | 38.90 | 0.966 |
| SWORD (Xu et al., 2024) | ✓ | 42.35 | 0.973 | 40.93 | 0.967 | 39.42 | 0.960 | 37.63 | 0.947 |
| FBP (Bracewell & Riddle, 1967) | ✗ | 23.09 | 0.515 | 19.42 | 0.462 | 16.89 | 0.416 | 14.02 | 0.355 |
| NeRP (Shen et al., 2022) | ✗ | 26.91 | 0.801 | 26.68 | 0.789 | 26.95 | 0.802 | 26.66 | 0.785 |
| GBIR (Ours) | ✗ | **43.43** | **0.982** | **42.24** | **0.980** | **40.82** | **0.976** | **40.11** | **0.975** |

Table 11: SV-CT reconstruction on **Rib Fracture**. Best in **Bold.**

| Method | Extra Data | 180-view | | 120-view | | 90-view | | 60-view | |
|---|---|---|---|---|---|---|---|---|---|
| | | PSNR | SSIM | PSNR | SSIM | PSNR | SSIM | PSNR | SSIM |
| RED-CNN (Chen et al., 2017) | ✓ | 29.61 | 0.707 | 28.97 | 0.682 | 27.94 | 0.658 | 27.97 | 0.585 |
| MCG (Chung et al., 2022) | ✓ | 34.81 | 0.851 | 34.94 | 0.852 | 34.96 | 0.853 | 35.07 | 0.854 |
| DiffusionMBIR (Chung et al., 2023) | ✓ | 34.64 | 0.950 | 34.64 | 0.952 | 34.54 | 0,951 | 34.35 | **0.950** |
| SWORD (Xu et al., 2024) | ✓ | 36.51 | 0.877 | 35.90 | 0.864 | 35.53 | 0.855 | 34.76 | 0.838 |
| FBP (Bracewell & Riddle, 1967) | ✗ | 19.33 | 0.388 | 16.64 | 0.324 | 14.76 | 0.280 | 12.69 | 0.230 |
| NeRP (Shen et al., 2022) | ✗ | 25.77 | 0.778 | 25.10 | 0.744 | 25.63 | 0.771 | 25.60 | 0.769 |
| GBIR (Ours) | ✗ | **42.43** | **0.972** | **41.05** | **0.962** | **40.01** | **0.953** | **39.43** | 0.948 |

Table 12: SV-CT reconstruction on **Appendicitis**. Best in **Bold.**

| Method | Extra Data | 180-view | | 120-view | | 90-view | | 60-view | |
|---|---|---|---|---|---|---|---|---|---|
| | | PSNR | SSIM | PSNR | SSIM | PSNR | SSIM | PSNR | SSIM |
| RED-CNN (Chen et al., 2017) | ✓ | 36.96 | 0.904 | 35.54 | 0.906 | 31.30 | 0.838 | 32.59 | 0.854 |
| MCG (Chung et al., 2022) | ✓ | 38.76 | 0.908 | 38.96 | 0.909 | 38.97 | 0.897 | 38.36 | 0.899 |
| DiffusionMBIR (Chung et al., 2023) | ✓ | 38.34 | 0.960 | 38.28 | 0.959 | 38.24 | 0.966 | 38.00 | 0.967 |
| SWORD (Xu et al., 2024) | ✓ | **44.18** | 0.976 | **42.62** | 0.971 | 40.85 | 0.964 | 37.79 | 0.949 |
| FBP (Bracewell & Riddle, 1967) | ✗ | 23.37 | 0.516 | 19.63 | 0.462 | 18.17 | 0.427 | 14.67 | 0.366 |
| NeRP (Shen et al., 2022) | ✗ | 27.15 | 0.821 | 27.25 | 0.817 | 27.38 | 0.819 | 27.28 | 0.817 |
| GBIR (Ours) | ✗ | 42.03 | **0.981** | 41.63 | **0.981** | 41.15 | **0.979** | 40.22 | **0.976** |

Table 13: SV-CT reconstruction on **Pneumonia**. Best in **Bold.**

| Method | Extra Data | 180-view | | 120-view | | 90-view | | 60-view | |
|---|---|---|---|---|---|---|---|---|---|
| | | PSNR | SSIM | PSNR | SSIM | PSNR | SSIM | PSNR | SSIM |
| RED-CNN (Chen et al., 2017) | ✓ | 31.78 | 0.733 | 30.43 | 0.672 | 29.22 | 0.680 | 27.82 | 0.578 |
| MCG (Chung et al., 2022) | ✓ | 32.87 | 0.810 | 33.05 | 0.813 | 33.19 | 0.814 | 33.33 | 0.815 |
| DiffusionMBIR (Chung et al., 2023) | ✓ | 33.34 | 0.954 | 33.26 | 0.953 | 33.10 | 0.952 | 32.86 | **0.951** |
| SWORD (Xu et al., 2024) | ✓ | 39.69 | 0.901 | 38.75 | 0.887 | 38.02 | 0.875 | 36.40 | 0.850 |
| FBP (Bracewell & Riddle, 1967) | ✗ | 17.57 | 0.323 | 15.73 | 0.264 | 14.66 | 0.229 | 12.73 | 0.182 |
| NeRP (Shen et al., 2022) | ✗ | 25.52 | 0.694 | 26.16 | 0.733 | 25.93 | 0.722 | 25.64 | 0.701 |
| GBIR (Ours) | ✗ | **41.77** | **0.967** | **40.96** | **0.962** | **40.31** | **0.956** | **39.11** | 0.946 |

Table 14: SV-CT reconstruction on **Cerebral Hemorrhage**. Best in **Bold.**

| Method | Extra Data | 180-view | | 120-view | | 90-view | | 60-view | |
|---|---|---|---|---|---|---|---|---|---|
| | | PSNR | SSIM | PSNR | SSIM | PSNR | SSIM | PSNR | SSIM |
| RED-CNN (Chen et al., 2017) | ✓ | 35.47 | 0.895 | 33.29 | 0.864 | 30.46 | 0.786 | 29.26 | 0.766 |
| MCG (Chung et al., 2022) | ✓ | 39.14 | 0.898 | 39.23 | 0.899 | 39.32 | 0.899 | 39.31 | 0.899 |
| DiffusionMBIR (Chung et al., 2023) | ✓ | 39.04 | 0.969 | 39.29 | 0.973 | 39.05 | 0.971 | 38.53 | 0.969 |
| SWORD (Xu et al., 2024) | ✓ | 34.90 | 0.742 | 33.50 | 0.740 | 31.86 | 0.737 | 29.57 | 0.732 |
| FBP (Bracewell & Riddle, 1967) | ✗ | 24.13 | 0.526 | 21.54 | 0.490 | 19.70 | 0.460 | 17.52 | 0.413 |
| NeRP (Shen et al., 2022) | ✗ | 25.38 | 0.789 | 25.98 | 0.804 | 25.02 | 0.760 | 24.23 | 0.764 |
| GBIR (Ours) | ✗ | **43.71** | **0.984** | **42.94** | **0.981** | **41.68** | **0.978** | **40.56** | **0.974** |

Table 15: SV-CT reconstruction on **Kidney Stones**. Best in **Bold.**

| Method | Extra Data | 180-view | | 120-view | | 90-view | | 60-view | |
|---|---|---|---|---|---|---|---|---|---|
| | | PSNR | SSIM | PSNR | SSIM | PSNR | SSIM | PSNR | SSIM |
| RED-CNN (Chen et al., 2017) | ✓ | 36.65 | 0.882 | 34.70 | 0.909 | 31.98 | 0.802 | 30.89 | 0.798 |
| MCG (Chung et al., 2022) | ✓ | 38.16 | 0.909 | 38.43 | 0.911 | 38.49 | 0.912 | 38.67 | 0.913 |
| DiffusionMBIR (Chung et al., 2023) | ✓ | 28.84 | 0.964 | 38.92 | 0.966 | 38.79 | 0.964 | 38.54 | 0.964 |
| SWORD (Xu et al., 2024) | ✓ | 43.58 | 0.980 | 42.27 | 0.976 | 40.95 | 0.971 | 39.51 | 0.961 |
| FBP (Bracewell & Riddle, 1967) | ✗ | 22.88 | 0.483 | 19.39 | 0.439 | 16.27 | 0.398 | 13.52 | 0.341 |
| NeRP (Shen et al., 2022) | ✗ | 26.17 | 0.767 | 26.25 | 0.773 | 26.11 | 0.772 | 26.16 | 0.776 |
| GBIR (Ours) | ✗ | **44.37** | **0.988** | **43.45** | **0.987** | **42.99** | **0.986** | **41.20** | **0.982** |

Table 16: SV-CT reconstruction on **Fatty Liver**. Best in **Bold.**

| Method | Extra Data | 180-view | | 120-view | | 90-view | | 60-view | |
|---|---|---|---|---|---|---|---|---|---|
| | | PSNR | SSIM | PSNR | SSIM | PSNR | SSIM | PSNR | SSIM |
| RED-CNN (Chen et al., 2017) | ✓ | 36.60 | 0.857 | 35.64 | 0.876 | 32.48 | 0.743 | 32.73 | 0.836 |
| MCG (Chung et al., 2022) | ✓ | 37.97 | 0.897 | 38.07 | 0.897 | 38.12 | 0.898 | 38.14 | 0.898 |
| DiffusionMBIR (Chung et al., 2023) | ✓ | 38.04 | 0.961 | 37.95 | 0.960 | 37.86 | 0.960 | 37.68 | 0.959 |
| SWORD (Xu et al., 2024) | ✓ | 43.47 | 0.973 | 42.21 | 0.968 | 40.76 | 0.961 | 38.45 | 0.948 |
| FBP (Bracewell & Riddle, 1967) | ✗ | 22.29 | 0.482 | 18.10 | 0.431 | 16.54 | 0.395 | 13.87 | 0.342 |
| NeRP (Shen et al., 2022) | ✗ | 26.89 | 0.785 | 27.27 | 0.808 | 26.81 | 0.784 | 26.93 | 0.792 |
| GBIR (Ours) | ✗ | **44.46** | **0.987** | **43.96** | **0.986** | **43.47** | **0.985** | **42.54** | **0.983** |

Table 17: SV-CT reconstruction on **Gallbladder Stones**. Best in **Bold.**

| Method | Extra Data | 180-view | | 120-view | | 90-view | | 60-view | |
|---|---|---|---|---|---|---|---|---|---|
| | | PSNR | SSIM | PSNR | SSIM | PSNR | SSIM | PSNR | SSIM |
| RED-CNN (Chen et al., 2017) | ✓ | 36.15 | 0.892 | 35.59 | 0.913 | 32.41 | 0.797 | 31.80 | 0.868 |
| MCG (Chung et al., 2022) | ✓ | 38.13 | 0.897 | 38.47 | 0.901 | 38.01 | 0.897 | 37.95 | 0.899 |
| DiffusionMBIR (Chung et al., 2023) | ✓ | 38.20 | 0.966 | 38.22 | 0.966 | 38.19 | 0.967 | 37.86 | 0.965 |
| SWORD (Xu et al., 2024) | ✓ | 43.66 | 0.974 | 42.34 | 0.969 | 40.56 | 0.961 | 37.64 | 0.943 |
| FBP (Bracewell & Riddle, 1967) | ✗ | 23.94 | 0.548 | 20.27 | 0.494 | 17.46 | 0.445 | 14.68 | 0.380 |
| NeRP (Shen et al., 2022) | ✗ | 27.03 | 0.809 | 27.12 | 0.814 | 26.81 | 0.799 | 26.86 | 0.806 |
| GBIR (Ours) | ✗ | **43.73** | **0.985** | **42.91** | **0.984** | **42.15** | **0.982** | **40.55** | **0.977** |

Table 18: SV-CT reconstruction on **Hepatic Cyst**. Best in **Bold.**

| Method | Extra Data | 180-view | | 120-view | | 90-view | | 60-view | |
|---|---|---|---|---|---|---|---|---|---|
| | | PSNR | SSIM | PSNR | SSIM | PSNR | SSIM | PSNR | SSIM |
| RED-CNN (Chen et al., 2017) | ✓ | 36.74 | 0.930 | 35.52 | 0.905 | 31.33 | 0.791 | 33.36 | 0.854 |
| MCG (Chung et al., 2022) | ✓ | 37.87 | 0.891 | 37.91 | 0.891 | 37.94 | 0.891 | 37.94 | 0.891 |
| DiffusionMBIR (Chung et al., 2023) | ✓ | 37.92 | 0.955 | 38.02 | 0.957 | 37.91 | 0.956 | 37.50 | 0.952 |
| SWORD (Xu et al., 2024) | ✓ | 42.84 | 0.973 | 41.42 | 0.967 | 39.81 | 0.960 | 37.12 | 0.946 |
| FBP (Bracewell & Riddle, 1967) | ✗ | 25.26 | 0.603 | 19.94 | 0.525 | 17.27 | 0.475 | 14.26 | 0.416 |
| NeRP (Shen et al., 2022) | ✗ | 26.65 | 0.808 | 26.57 | 0.804 | 26.65 | 0.808 | 26.39 | 0.799 |
| GBIR (Ours) | ✗ | **42.96** | **0.981** | **42.29** | **0.980** | **41.47** | **0.977** | **39.12** | **0.971** |

Table 19: SV-CT reconstruction on **Elbow Fracture**. Best in **Bold.**

| Method | Extra Data | 180-view | | 120-view | | 90-view | | 60-view | |
|---|---|---|---|---|---|---|---|---|---|
| | | PSNR | SSIM | PSNR | SSIM | PSNR | SSIM | PSNR | SSIM |
| RED-CNN (Chen et al., 2017) | ✓ | 34.41 | 0.847 | 34.05 | 0.789 | 27.42 | 0.777 | 29.82 | 0.732 |
| MCG (Chung et al., 2022) | ✓ | 37.20 | 0.857 | 37.13 | 0.858 | 37.08 | 0.856 | 36.80 | 0.852 |
| DiffusionMBIR (Chung et al., 2023) | ✓ | 37.06 | 0.932 | 36.93 | 0.930 | 36.89 | 0.931 | 36.75 | 0.930 |
| SWORD (Xu et al., 2024) | ✓ | **42.83** | 0.959 | 38.67 | 0.917 | 37.39 | 0.901 | 34.71 | 0.865 |
| FBP (Bracewell & Riddle, 1967) | ✗ | 26.15 | 0.459 | 22.32 | 0.382 | 19.93 | 0.337 | 16.95 | 0.279 |
| NeRP (Shen et al., 2022) | ✗ | 28.14 | 0.826 | 28.31 | 0.827 | 28.06 | 0.823 | 28.18 | 0.835 |
| GBIR (Ours) | ✗ | 42.82 | **0.961** | **41.94** | **0.954** | **41.19** | **0.949** | **38.97** | **0.978** |

Table 20: SV-CT reconstruction on **Spinal Fracture**. Best in **Bold.**

| Method | Extra Data | 180-view | | 120-view | | 90-view | | 60-view | |
|---|---|---|---|---|---|---|---|---|---|
| | | PSNR | SSIM | PSNR | SSIM | PSNR | SSIM | PSNR | SSIM |
| RED-CNN (Chen et al., 2017) | ✓ | 23.86 | 0.866 | 23.94 | 0.841 | 23.92 | 0.832 | 23.70 | 0.810 |
| MCG (Chung et al., 2022) | ✓ | 38.52 | 0.913 | 38.52 | 0.913 | 38.48 | 0.912 | 38.40 | 0.911 |
| DiffusionMBIR (Chung et al., 2023) | ✓ | 39.34 | 0.973 | 39.27 | 0.973 | **39.08** | **0.972** | **38.49** | **0.969** |
| SWORD (Xu et al., 2024) | ✓ | 40.94 | 0.946 | 38.02 | 0.930 | 34.68 | 0.901 | 28.85 | 0.834 |
| FBP (Bracewell & Riddle, 1967) | ✗ | 16.41 | 0.793 | 15.20 | 0.766 | 14.73 | 0.741 | 13.96 | 0.698 |
| NeRP (Shen et al., 2022) | ✗ | 28.10 | 0.847 | 26.24 | 0.779 | 27.95 | 0.840 | 26.48 | 0.790 |
| GBIR (Ours) | ✗ | **41.23** | **0.981** | **39.70** | **0.977** | 38.41 | 0.971 | 37.68 | 0.968 |

Table 21: SV-CT reconstruction on **Foot Fracture**. Best in **Bold**.

| Method | Extra Data | 180-view | | 120-view | | 90-view | | 60-view | |
|---|---|---|---|---|---|---|---|---|---|
| | | PSNR | SSIM | PSNR | SSIM | PSNR | SSIM | PSNR | SSIM |
| RED-CNN (Chen et al., 2017) | ✓ | 37.53 | 0.860 | 35.61 | 0.783 | 32.52 | 0.817 | 32.46 | 0.837 |
| MCG (Chung et al., 2022) | ✓ | 39.40 | 0.891 | 39.62 | 0.895 | 39.43 | 0.894 | 39.45 | 0.894 |
| DiffusionMBIR (Chung et al., 2023) | ✓ | 40.45 | 0.956 | 40.31 | 0.955 | 40.22 | 0.954 | **40.26** | 0.957 |
| SWORD (Xu et al., 2024) | ✓ | 34.92 | 0.927 | 36.40 | 0.905 | 31.95 | 0.866 | 28.33 | 0.783 |
| FBP (Bracewell & Riddle, 1967) | ✗ | 23.45 | 0.235 | 18.80 | 0.181 | 17.23 | 0.160 | 14.46 | 0.132 |
| NeRP (Shen et al., 2022) | ✗ | 30.69 | 0.921 | 30.82 | 0.926 | 30.76 | 0.932 | 30.56 | 0.927 |
| GBIR (Ours) | ✗ | **41.81** | **0.981** | **41.21** | **0.980** | **40.51** | **0.974** | 39.60 | **0.977** |

Table 22: SV-CT reconstruction on **Wrist Fracture**. Best in **Bold**.

| Method | Extra Data | 180-view | | 120-view | | 90-view | | 60-view | |
|---|---|---|---|---|---|---|---|---|---|
| | | PSNR | SSIM | PSNR | SSIM | PSNR | SSIM | PSNR | SSIM |
| RED-CNN (Chen et al., 2017) | ✓ | 36.61 | 0.810 | 34.73 | 0.825 | 31.73 | 0.870 | 30.78 | 0.744 |
| MCG (Chung et al., 2022) | ✓ | 37.14 | 0.887 | 37.53 | 0.889 | 37.65 | 0.890 | 37.64 | 0.889 |
| DiffusionMBIR (Chung et al., 2023) | ✓ | 36.91 | 0.953 | 36.94 | 0.954 | 36.73 | 0.952 | 36.31 | 0.950 |
| SWORD (Xu et al., 2024) | ✓ | 36.74 | 0.903 | 33.91 | 0.874 | 31.57 | 0.832 | 28.91 | 0.766 |
| FBP (Bracewell & Riddle, 1967) | ✗ | 21.35 | 0.231 | 17.95 | 0.197 | 15.69 | 0.174 | 12.96 | 0.146 |
| NeRP (Shen et al., 2022) | ✗ | 29.55 | 0.893 | 28.77 | 0.891 | 29.62 | 0.897 | 29.56 | 0.893 |
| GBIR (Ours) | ✗ | **40.28** | **0.984** | **39.71** | **0.983** | **38.89** | **0.976** | **37.78** | **0.973** |

Table 23: SV-CT reconstruction on **Subarachnoid Hemorrhage**. Best in **Bold**.

| Method | Extra Data | 180-view | | 120-view | | 90-view | | 60-view | |
|---|---|---|---|---|---|---|---|---|---|
| | | PSNR | SSIM | PSNR | SSIM | PSNR | SSIM | PSNR | SSIM |
| RED-CNN (Chen et al., 2017) | ✓ | 29.52 | 0.874 | 29.65 | 0.877 | 29.55 | 0.877 | 29.42 | 0.863 |
| MCG (Chung et al., 2022) | ✓ | 38.78 | 0.908 | 38.87 | 0.909 | 38.81 | 0.908 | 38.79 | 0.908 |
| DiffusionMBIR (Chung et al., 2023) | ✓ | 39.46 | 0.975 | 39.38 | 0.975 | 39.20 | 0.974 | 38.70 | 0.973 |
| SWORD (Xu et al., 2024) | ✓ | 42.54 | 0.965 | 39.60 | 0.955 | 36.71 | 0.938 | 31.84 | 0.895 |
| FBP (Bracewell & Riddle, 1967) | ✗ | 21.31 | 0.440 | 20.84 | 0.423 | 19.22 | 0.404 | 17.93 | 0.361 |
| NeRP (Shen et al., 2022) | ✗ | 23.72 | 0.760 | 23.34 | 0.760 | 23.84 | 0.800 | 24.04 | 0.791 |
| GBIR (Ours) | ✗ | **43.29** | **0.993** | **42.74** | **0.993** | **41.98** | **0.992** | **41.02** | **0.992** |

Table 24: CS-MRI reconstruction on **Brain**. Best in **Bold**.

| Method | Extra Data | Axial | | Coronal | | Sagittal | |
|---|---|---|---|---|---|---|---|
| | | PSNR | SSIM | PSNR | SSIM | PSNR | SSIM |
| RED-CNN (Chen et al., 2017) | ✓ | 26.36 | 0.686 | 29.49 | 0.786 | 29.05 | 0.731 |
| AUTOMAP (Zhu et al., 2018) | ✓ | 19.06 | 0.635 | 17.75 | 0.593 | 17.58 | 0.489 |
| ScoreMRI (Chung & Ye, 2022) | ✓ | 25.17 | 0.725 | **32.46** | 0.786 | 28.99 | 0.763 |
| DiffusionMBIR (Chung et al., 2023) | ✓ | 22.37 | 0.703 | 26.13 | 0.694 | **29.13** | 0.784 |
| IFFT (Gallagher et al., 2008) | ✗ | 21.39 | 0.711 | 20.60 | 0.750 | 20.96 | 0.726 |
| GBIR (Ours) | ✗ | **28.89** | **0.895** | 29.21 | **0.914** | 29.07 | **0.937** |

Table 25: CS-MRI reconstruction on **Abdomen**. Best in **Bold.**

| Method | Extra Data | Axial | | Coronal | | Sagittal | |
|---|---|---|---|---|---|---|---|
| | | PSNR | SSIM | PSNR | SSIM | PSNR | SSIM |
| RED-CNN (Chen et al., 2017) | ✓ | 33.63 | 0.858 | 31.48 | 0.847 | 31.39 | 0.671 |
| AUTOMAP (Zhu et al., 2018) | ✓ | 15.64 | 0.514 | 17.20 | 0.516 | 17.61 | 0.397 |
| ScoreMRI (Chung & Ye, 2022) | ✓ | 31.26 | 0.789 | 31.34 | 0.839 | 28.09 | 0.604 |
| DiffusionMBIR (Chung et al., 2023) | ✓ | 28.45 | 0.731 | 21.72 | 0.793 | 25.69 | 0.654 |
| IFFT (Gallagher et al., 2008) | ✗ | 19.23 | 0.718 | 19.93 | 0.750 | 22.29 | 0.665 |
| GBIR (Ours) | ✗ | **35.15** | **0.935** | **33.29** | **0.922** | **34.86** | **0.955** |

Table 26: CS-MRI reconstruction on **Shoulder**. Best in **Bold.**

| Method | Extra Data | Axial | | Coronal | | Sagittal | |
|---|---|---|---|---|---|---|---|
| | | PSNR | SSIM | PSNR | SSIM | PSNR | SSIM |
| RED-CNN (Chen et al., 2017) | ✓ | 27.95 | 0.724 | 27.41 | 0.769 | 29.64 | 0.746 |
| AUTOMAP (Zhu et al., 2018) | ✓ | 21.81 | 0.657 | 23.83 | 0.622 | 21.18 | 0.663 |
| ScoreMRI (Chung & Ye, 2022) | ✓ | 29.12 | 0.763 | 30.89 | **0.873** | 30.55 | 0.814 |
| DiffusionMBIR (Chung et al., 2023) | ✓ | 30.01 | **0.781** | 27.90 | 0.788 | 30.74 | 0.863 |
| IFFT (Gallagher et al., 2008) | ✗ | 28.66 | 0.733 | 27.49 | 0.693 | 28.83 | 0.745 |
| GBIR (Ours) | ✗ | **32.64** | 0.730 | **31.71** | 0.868 | **33.54** | **0.919** |

Table 27: CS-MRI reconstruction on **Knee**. Best in **Bold.**

| Method | Extra Data | Axial | | Coronal | | Sagittal | |
|---|---|---|---|---|---|---|---|
| | | PSNR | SSIM | PSNR | SSIM | PSNR | SSIM |
| RED-CNN (Chen et al., 2017) | ✓ | 32.98 | 0.786 | 30.29 | **0.862** | 27.86 | 0.822 |
| AUTOMAP (Zhu et al., 2018) | ✓ | 24.07 | 0.818 | 22.31 | 0.667 | 18.22 | 0.654 |
| ScoreMRI (Chung & Ye, 2022) | ✓ | 30.75 | 0.623 | **33.18** | 0.847 | **31.46** | 0.780 |
| DiffusionMBIR (Chung et al., 2023) | ✓ | 27.63 | 0.617 | 29.46 | 0.813 | 23.94 | 0.760 |
| IFFT (Gallagher et al., 2008) | ✗ | 29.98 | **0.827** | 26.30 | 0.790 | 20.48 | 0.713 |
| GBIR (Ours) | ✗ | **34.11** | 0.753 | 29.70 | 0.826 | 27.13 | **0.857** |

Table 28: CS-MRI reconstruction on **Elbow**. Best in **Bold.**

| Method | Extra Data | Axial | | Coronal | | Sagittal | |
|---|---|---|---|---|---|---|---|
| | | PSNR | SSIM | PSNR | SSIM | PSNR | SSIM |
| RED-CNN (Chen et al., 2017) | ✓ | 27.17 | **0.846** | 30.34 | 0.673 | 27.63 | 0.810 |
| AUTOMAP (Zhu et al., 2018) | ✓ | 16.30 | 0.521 | 20.97 | 0.701 | 16.23 | 0.458 |
| ScoreMRI (Chung & Ye, 2022) | ✓ | **31.54** | 0.814 | 29.58 | 0.580 | 28.72 | 0.813 |
| DiffusionMBIR (Chung et al., 2023) | ✓ | 29.76 | 0.823 | 30.11 | 0.732 | 29.45 | 0.798 |
| IFFT (Gallagher et al., 2008) | ✗ | 23.34 | 0.810 | 22.86 | 0.662 | 23.14 | 0.771 |
| GBIR (Ours) | ✗ | 30.05 | 0.812 | **31.08** | **0.878** | **29.96** | **0.897** |

