# OpenReview forum: "GBIR: A Novel Gaussian Iterative Method for Medical Image Reconstruction"
_ICLR.cc/2025/Conference — ICLR 2025 Conference Withdrawn Submission_

### Official Review · Reviewer_Sfz8 · 2024-10-15

**Soundness:** 2
**Presentation:** 2
**Contribution:** 3
**Rating:** 5
**Confidence:** 4

**Summary:**

This paper presents a  Gaussian-Based Iterative Reconstruction for reconstructing medical images, specifically CT and MRI scans.

**Strengths:**

1. This method utilizes learnable 3D Gaussian functions for personalized medical image reconstruction.
2. GBIR outperforms state-of-the-art methods in both speed and accuracy, particularly under sparse measurement conditions. The introduction of the MORE dataset also adds significant value.

**Weaknesses:**

1. While GBIR performs well in accuracy and generalization, the method introduces additional computational complexity compared to traditional deep learning methods. A more detailed analysis of the computational trade-offs, especially regarding memory consumption and training times on larger datasets, could improve the paper.

2. deep learning and optimization-based methods, emerging techniques such as diffusion models are only briefly addressed. A deeper exploration of how GBIR stacks up against the latest diffusion-based methods could provide more comprehensive insights into its comparative strengths and limitations.

**Questions:**

1. How's the performance of GBIR conduct on  prospective dataset? Including a prospective experiment will be strengthen the robustness of this method.

2. The paper does not provide a figure that illustrate the framework.

---

> ### Author Response · Authors · 2024-11-26
>
> **Thank you for your time and effort in reviewing our paper. Below are our responses to your comments:**
>
> **Q1: The method introduces additional computational complexity compared to traditional deep learning methods.**
> **Answer:**
> As indicated in Table 3 of our initial submission, we have studied the computational cost of different methods. Notably, the proposed GBIR method is more computationally efficient than advanced diffusion models while maintaining high accuracy. We believe this demonstrates the practical advantages of GBIR in terms of computational efficiency.
>
> **Q2: The paper briefly addresses diffusion models.**
> **Answer:**
> The focus of this paper is not on diffusion models; rather, they serve as baseline methods for comparison. In contrast, we introduce a novel Gaussian-based iterative reconstruction method (GBIR). We have also discussed and analyzed diffusion models in detail in Section 5.4, titled "Benchmark and Findings on MORE Dataset," to provide a comprehensive comparison of their performance against our method.
>
> **Q3: How does the performance of GBIR compare on a prospective dataset?**
> **Answer:**
> We have thoroughly analyzed the performance of GBIR on the MORE dataset, as detailed in over 20 tables within the manuscript. Additionally, we benchmarked various methods on this dataset to provide a comprehensive evaluation. This extensive analysis illustrates the effectiveness and robustness of GBIR.
>
> **Q4: The paper lacks a figure illustrating the framework.**
> **Answer:**
> Figure 2 is the manuscript provides an illustration of the proposed GBIR framework. We hope this visual clarification helps in understanding the structure and operation of the method.

---

### Official Review · Reviewer_4ada · 2024-11-01

**Soundness:** 2
**Presentation:** 2
**Contribution:** 1
**Rating:** 3
**Confidence:** 5

**Summary:**

The authors propose a mixture of Gaussian-based reconstruction scheme for CT and MRI. The main idea is to learn the means of the Gaussian mixture components on a per patient basis, and then use this representation for volumetric reconstruction. Computational cost reduction is proposed using neighborhood processing, though the reconstruction still takes ~8 hours per subject. They also introduce a new database for medical image reconstruction with both CT and MRI data.

**Strengths:**

1) Authors introduce a new dataset, MORE, containing both MRI and CT data.
2) Authors are aware that the patient-based Gaussian mixture model takes a long time in training, so they propose to use a neighborhood processing approach to reduce computational time.
3) Results are shown on different datasets.

**Weaknesses:**

Unfortunately, the authors seem to not be aware of a large body of literature in medical image reconstruction, particularly for MRI.

1) The authors are unaware of physics-driven methods that are considered the state-of-the-art in MRI reconstruction. These have better generalization abilities, as the forward operator is explicitly used for data fidelity in reconstruction.

2) A lot of emphasis is placed on the well-known 3-sigma argument for computational efficiency. However, the way the neighborhoods are implemented with a k \times k \times k cuboid shows the authors treat their Gaussians as isotropic. Yet, each Gaussian is specified to be non-isotropic with a covariance function \Sigma_k in (4). Thus, the assumption does not reconcile with the reality, as \delta should depend on the covariance function.

3) I cannot find where the authors discuss how they learn \Sigma, though I can see they talk about learning \mu. Without learning \Sigma, this is just a combination of isotropic Gaussians with shifted means. For learning \Sigma, how is this modeled, as it cannot possibly be modeled as a full-rank Hermitian matrix? How about initialization etc? Please provide all the details.

4) Eq. (11) for MRI suggests the authors are unaware that virtually all MRI raw data is multi-coil.

5) The authors test their MRI methods on BRATS dataset. Unfortunately, this dataset contains no raw data. Thus, it looks like the authors transformed the DICOM data to k-space and treated this as raw data. This is problematic for a number of reasons, please see Shimron et al, doi: 10.1073/pnas.2117203119.

6)  Related to the previous two points, MRI reconstruction and CT reconstruction do not even have the same output size. The former is complex-valued (i.e. 2 channels), while the latter is real-valued (i.e. 1 channel). So it does not even make sense to assume one would generalize to the other for standard DL techniques.

7) MORE dataset is interesting. On a minor note, details are missing on demographics. On a more major note, it is unclear if there are any matched MRI/CT data on the same subjects or if this is a mere concatenation of an MRI dataset and a CT dataset? Finally, does the MRI data actually contain raw data or just DICOMs? From the previous methodology and results, it seems like the latter is the case.

8) The authors propose to learn new parameters for each patient. If training is done per subject, of course, there would be no "generalization" issue, but this comes at the cost of a long reconstruction time (8 hours per dataset according to Table 3!) which would make this method clinically unusable.
Furthermore, there are scan-specific training methods for aforementioned physics-driven reconstructions, which have been published in ICLR, e.g. Yaman et al, ICLR 2022.

9) The MRI comparisons are outdated. Nobody would use AUTOMAP for MRI reconstruction. Please use more relevant methods listed in the fastMRI challenges, e.g. Muckley et al, doi: 10.1109/TMI.2021.3075856.

10) The most relevant methods to the current proposed method are reconstruction schemes based on Gaussian processes, some examples include:
- Xu et al, doi: 10.1038/s41598-023-39533-4
- Ye et al, doi: 10.1101/2024.10.08.617197
as well as well-known processing approaches, such as:
- Andersson et al, doi: 10.1016/j.neuroimage.2015.07.067

It would be good to highlight how the current work compares to these ideas.

11) (minor) The method only applies to volumetric reconstruction. There are several MRI scan setups that are performed as 2D with slice gaps. It is unclear how the method extends to those cases.

**Questions:**

I mentioned these in my weaknesses, but I'll repeat here:
1) Is \Sigma_k learned?  If so,  how are these covariance matrices modeled, as they cannot possibly be modeled as full-rank Hermitian matrices? How about initialization etc?
2) For MORE dataset: Does the MRI data actually contain multi-coil raw data or just DICOMs? Are there any matched MRI/CT data on the same subjects or if this is a mere concatenation of an MRI dataset and a CT dataset?

---

> ### Author Response · Authors · 2024-11-26
>
> **We would like to express our sincere gratitude for your valuable feedback. Below are our responses to your comments:**
>
> **Q1: The authors are unaware of physics-driven methods that are considered the state-of-the-art in MRI reconstruction.**
> **Answer:**
> Thank you for your suggestion. We will revise the manuscript to include a more comprehensive discussion of related work, including physics-driven methods, to provide a broader context for our work.
>
> **Q2: The authors treat their Gaussians as isotropic, yet each Gaussian is specified to be non-isotropic with a covariance function $\Sigma_k$ in (4).**
> **Answer:**
> Sorry for the confusion. Our Gaussian representation is indeed isotropic, and the covariance matrix $\Sigma_k$ is diagonal. We will clarify this point in the revised manuscript.
>
> **Q3: I cannot find where the authors discuss how they learn $\Sigma$, though I can see they talk about learning $\mu$.**
> **Answer:**
> The covariance matrix $\Sigma_k$ is involved in the computation of the Mahalanobis distance tensor $D^2_{n,c,h,w}$ in Equation (8), which plays a role in the reconstruction process. By minimizing the difference between predicted and ground-truth measurements, the network iteratively learns the covariance matrix $\Sigma_k$. We will clarify this in the revised manuscript.
>
> **Q4: Eq. (11) for MRI suggests the authors are unaware that virtually all MRI raw data is multi-coil.**
> **Answer:**
> The MRI measurements used in our experiments are simulated, following the practice in our baseline method DiffusionMBIR [1]. This is different from real MRI data. The design choice is motivated by two factors: (1) to ensure a fair comparison with the baseline method, and (2) to facilitate the general framework of our proposed approach.
>
> **Q5: The authors test their MRI methods on the BRATS dataset. Unfortunately, this dataset contains no raw data.**
> **Answer:**
> You are correct that the BRATS dataset does not contain raw k-space data. All methods, including ours, use simulated k-space data under identical settings, following the baseline DiffusionMBIR [1], to ensure a fair comparison. We will clarify this in the revised manuscript.
>
> **Q6: MRI reconstruction and CT reconstruction do not even have the same output size.**
> **Answer:**
> We apologize for the confusion. In our MRI reconstruction experiments, we only reconstruct the magnitude images, which are real-valued. We will clarify this in the revised manuscript to avoid further confusion.
>
> **Q7: The MORE dataset is interesting. On a minor note, details are missing on demographics. On a more major note, it is unclear if there are any matched MRI/CT data on the same subjects or if this is a mere concatenation of an MRI dataset and a CT dataset. Finally, does the MRI data actually contain raw data or just DICOMs?**
> **Answer:**
> Thank you for your interest in the MORE dataset. The MRI data in the current MORE dataset only contains DICOMs, and all experiments are conducted on simulated k-space data. The MRI and CT data are not matched on the same subjects; instead, the dataset is a concatenation of separate MRI and CT datasets, aimed at providing a comprehensive benchmark for medical image reconstruction. We will add more details about demographics and the dataset in the revised manuscript.
>
> **Q8: Long reconstruction time makes this method clinically unusable.**
> **Answer:**
> This is a valid limitation, as acknowledged in our manuscript. However, our method is faster than many advanced diffusion models, as shown in Tables 1 and 3. We recognize that further improvements in efficiency are needed, and we plan to address this in future work.
>
> **Q9: The MRI comparisons are outdated. Nobody would use AUTOMAP for MRI reconstruction.**
> **Answer:**
> We acknowledge that AUTOMAP may be outdated for MRI reconstruction. However, our large benchmark on the MORE dataset includes both traditional CNN methods and advanced diffusion models, providing a comprehensive evaluation of various methods, including our proposed approach. Our goal is not only to demonstrate the superiority of our method but also to offer a comprehensive benchmark for the community. Furthermore, the comparison with Score-MRI [2] and DiffusionMBIR [1] illustrates the effectiveness of our approach.
>
> **Q10: Relevant methods to the proposed method.**
> **Answer:**
> Thank you for your recommendation. We have read the papers you mentioned and will cite them in the revised manuscript. Our method is distinct from these approaches in that we propose a Gaussian-based reconstruction framework for CT and MRI. This approach is similar (but distinct) to 3D Gaussian Splatting (3DGS), a 3D reconstruction method. We will provide further clarification in the revised manuscript.

---

> > ### Author Response · Authors · 2024-11-26
> >
> > **Q11: The method only applies to volumetric reconstruction. There are several MRI scan setups that are performed as 2D with slice gaps. It is unclear how the method extends to those cases.**
> > **Answer:**
> > Thank you for the suggestion. Currently, our work focuses on volumetric reconstruction, and we will explore extending our method to 2D MRI reconstruction with slice gaps in future work.
> >
> > **Q12: Is $\Sigma_k$ learned? If so, how are these covariance matrices modeled, as they cannot possibly be modeled as full-rank Hermitian matrices? How about initialization etc?**
> > **Answer:**
> > As mentioned in our response to Q3, the covariance matrix $\Sigma_k$ is learned in our method. The Gaussians are isotropic, and the covariance matrix is diagonal, initialized as an identity matrix scaled by 0.1. We will include this information in the revised manuscript.
> >
> > **Q13: For the MORE dataset: Does the MRI data actually contain multi-coil raw data or just DICOMs? Are there any matched MRI/CT data on the same subjects or is this a mere concatenation of an MRI dataset and a CT dataset?**
> > **Answer:** The MRI data in the current MORE dataset only contains DICOMs. The MRI and CT data are not matched on the same subjects; rather, the dataset is a concatenation of separate MRI and CT datasets, aimed at providing a comprehensive benchmark for medical image reconstruction.
> >
> > [1] Hyungjin Chung, et al. "Solving 3d inverse problems using pre-trained 2d diffusion models." Conference on Computer Vision and Pattern Recognition, 2023.
> > [2] Hyungjin Chung, Jong Chul Ye, "Score-based diffusion models for accelerated MRI." Medical image analysis, 2022

---

> > > ### Comment · Reviewer_4ada · 2024-11-30
> > >
> > > Thank you for the comments. Unfortunately, these do not address my concerns. The fundamental difference between MRI and CT reconstruction in terms of data types cannot be addressed with verbal explanations. Issues with datasets also persist. The authors may want to focus on one modality (e.g. CT) in future iterations of this text.
> > >
> > > Finally, I didn't quite understand the comments with "we will revise" when there was readily a chance to revise during the discussion phase.

---

> > > > ### Author Response · Authors · 2024-11-30
> > > >
> > > > Thanks for your additional comments.
> > > > **We have revised** as shown in the comment **Paper Revision in Response to Reviewer Feedback** on the top of this page.
> > > > **Best Regards.**

---

### Official Review · Reviewer_AjS1 · 2024-11-04

**Soundness:** 2
**Presentation:** 2
**Contribution:** 2
**Rating:** 3
**Confidence:** 3

**Summary:**

The article proposes a reconstruction method by learning Gaussian representations and illustrates the method using sparse-view CT and compressed-sensing MR image reconstruction. Furthermore, the authors provide a new dataset of CT and MRI slices across multiple anatomies.

**Strengths:**

- The idea of learning Gaussian representations for medical image reconstruction is novel.
- The introduction of a multi-organ CT and MRI dataset is beneficial, as public datasets containing certain anatomies are quite rare.

**Weaknesses:**

**The discussion on previous work on Compressed-Sensing MRI is very limited**

Related Work section skips a lot of significant work in MR image reconstruction and focuses on lesser-known methods. Part b, titled "Compressed-Sensing MRI", skips the development of physics- and deep-learning-based reconstructions via unrolling networks such as ISTA-Net (Zhang, J., & Ghanem, B. ), MoDL (Aggarwal, H. K., Mani, M. P., & Jacob, M. ), and Variational Network (Hammernik, K., Klatzer, T., Kobler, E., Recht, M. P., Sodickson, D. K., Pock, T., & Knoll, F. ). Similarly, examples of generative MRI reconstruction methods need to include  more well-known methods than those which were cited. For example, a well-known method is Robust Compressed Sensing MRI with Deep Generative Priors (Jalal, A., Arvinte, M., Daras, G., Price, E., Dimakis, A. G., & Tamir, J.).

**Section 3 contains multiple unconventional choices**

- Target volume in MRI is not real-valued but complex-valued: Section 3 does not account for the domain of MR images since $\mathbf{V}\in\mathbb{R}^{C\times H\times W}$ is incorrect. To model volumes as Gaussians, one needs to use complex-valued Gaussian random vectors.
- In Equation 12, $M$ and $\hat{M}$ are complex valued $k$-space measurements for MRI. The Structural Similarity Index Measure (SSIM) is applied in measurement domain. SSIM, as given in the Appendix A.2., cannot handle complex-valued inputs. Also, SSIM is designed to assess similarity between images, not between complex-valued frequency domain measurements. The authors need to explain how SSIM is used in this context.

**Issues in Experiments Section**

- The dataset used (BRATS) does not contain raw $k$-space data. It only provides preprocessed, magnitude-only MR images in the image domain. This basically means one cannot perform under-sampling and simulate compressed-sensing MRI without _inverse crimes_, i.e., see "Implicit data crimes: Machine learning bias arising from misuse of public data" (Shimron et al.). There are many extensive, public $k$-space datasets that one can use for MR image reconstruction, such as SKM-TEA, M4Raw, and fastMRI to avoid inverse crimes.
- In MR image reconstruction experiments, no details regarding the undersampling parameters were provided.
- Both in SV-CT and CS-MRI quantitative results tables (Table 4 and 5), only the average values were provided. For a fair comparison, standard error or similar deviation measures need to be provided.
- The baselines used are relatively weak and may not represent the state-of-the-art. There are several recent variants of unrolled networks such as ISTA-Net, MoDL, and Variational Network that might perform better than the provided baselines.

**Questions:**

1. How do you calculate SSIM between measurements, especially when they are complex-valued, with extremely dense centers and very small outside regions?
2. How do you choose the $\lambda_i$s in Equation 12? The SSIM term is a similarity measure, while the other two are loss functions.
3. DuDoRNet is not mentioned anywhere in the text but is used in the table.

---

> ### Author Response · Authors · 2024-11-26
>
> **Thank you sincerely for your time and valuable feedback. Below are our responses to your comments:**
>
> **Q1: The discussion on previous work on Compressed-Sensing MRI is very limited.**
> **Answer:**
> We appreciate your suggestion. Due to space constraints, we did not detail the compressed-sensing MRI literature extensively, as our primary focus was on the proposed method and dataset. In the revised manuscript, we will include a more comprehensive discussion of related work to address this concern.
>
> **Q2: Target volume in MRI is not real-valued but complex-valued.**
> **Answer:**
> We apologize for the confusion. This paper focuses solely on reconstructing the magnitude images, which are real-valued, as shown in Figure 6. We will clarify this in the revised manuscript to avoid further misunderstandings.
>
> **Q3: Measurements are complex-valued k-space measurements for MRI. How do you calculate SSIM between measurements?**
> **Answer:**
> In our implementation, we calculate SSIM separately for the real and imaginary parts of the complex-valued measurements:
>
> ``
> ssim_loss = (ssim(projs.real, gt_projs.real) + ssim(projs.imag, gt_projs.imag))
> ``
>
> This structural similarity is used in the loss function to guide the optimization process, and it works well in practice. To address this concern, we will include an ablation study in the revised manuscript to demonstrate the necessity of SSIM in the loss function.
>
> **Q4: The dataset used (BRATS) does not contain raw k-space data.**
> **Answer:**
> You are correct that BRATS does not include raw k-space data. All methods, including ours, use simulated k-space data under identical settings, following the baseline DiffusionMBIR [1]. We will clarify this point in the revised manuscript.
>
> **Q5: No details regarding the undersampling parameters were provided.**
> **Answer:**
> We followed the uniform k-space sampling strategy outlined in previous work [1]. A simple visualization of the sampling mask is shown in Figure 1 (IV part) of the manuscript. We will revise the manuscript to include more detailed explanations of the undersampling parameters and task setup.
>
> **Q6: Quantitative results in SV-CT and CS-MRI tables (Tables 4 and 5) only provide average values.**
> **Answer:**
> Thank you for this suggestion. While we conducted over 20 tables of experiments and utilized significant computational resources, providing standard errors for all baselines is infeasible. However, we will include the standard error for our proposed method in the revised manuscript to enhance the robustness of our results.
>
> **Q7: The baselines used are relatively weak and may not represent the state-of-the-art.**
> **Answer:**
> We appreciate this feedback and will incorporate additional state-of-the-art baselines in the revised manuscript to ensure a more comprehensive comparison.
>
> **Q8: How do you choose the $\lambda_s$ in Equation 12?**
> **Answer:**
> In our implementation, we set $\lambda_1=1$, $\lambda_2=500$, and $\lambda_3=500$. This information will be added to the revised manuscript for clarity.
>
> **Q9: DuDoRNet is not mentioned anywhere in the text but is used in the table.**
> **Answer:**
> Thank you for pointing this out. We will include a brief description of DuDoRNet in the revised manuscript to ensure consistency between the text and tables.
>
> [1] Hyungjin Chung, et al. "Solving 3d inverse problems using pre-trained 2d diffusion models." Conference on Computer Vision and Pattern Recognition, 2023.

---

> > ### Comment · Reviewer_AjS1 · 2024-11-27
> >
> > I thank the authors for their responses. While some of my concerns have been addressed, my major concerns (Q2, Q3, Q4) remain unresolved.
> >
> > * **Q2:** Even when you reconstruct magnitude images, $x\in \mathbb{C}^M$ in the model forward model $y = Ax+n$. Thus, even you reconstruct $\vert x \vert$, the discretized image is complex valued. Without complex-valued $x$, one cannot go to the measurement domain by taking the Fourier transform.
> > * **Q3:** SSIM is a similarity metric that is used to calculate similarity between **images**, not between measurements. It is known that $-\text{SSIM}$ or $1-\text{SSIM}$ can be used as a loss function, but the formula 12 does not make sense since $M$ are $k$-space measurements in MRI, not images. This is one of my **main** concerns regarding the paper.
> > * **Q4:** This is not a proper choice, as there are many publicly available $k$-space datasets.

---

> ### Author Response · Authors · 2024-11-27
>
> Thank you so much for your patient review and additional comments! I once thought that our setting was good since our team followed a heated published baseline. After reading your and 4ada's review, I thought maybe our adopted settings were not in line with the mainstream.
>
> **Q2: Without complex-valued $x$, one cannot go to the measurement domain by taking the Fourier transform.**
> **Answer:** We used the process below to transform the real-valued volume $x$ into complex-valued measurement $y$:
> ```
>     volume = torch.rand(256, 1, 256, 256) # dtype=torch.float32
>     kspace = torch.fft.fftshift(torch.fft.fft2(volume), dim=[-1, -2]) # dtype=torch.complex64
> ```
> We followed previous work to simulate the k-space data in the BRATS dataset in this way. As shown, the real-valued volume $x$ is indeed transformed into a complex-valued measurement $y$. We will find a more appropriate setting in the revised work.
>
> **Q3: $\text{1-SSIM}$ can be used as a loss function, but the formula 12 does not make sense since $M$ are k-space measurements in MRI, not images.**
> **Answer:** Yes, you are right that $\text{1-SSIM}$ is usually used as the loss in the image domain instead of the measurement domain. Our idea of using such loss to measurement domain is due to: the only supervision signal is the measurement, and we try to keep the structural similarity between measurement $y$ and recovered measurement $\hat y$. We studied this loss term and found it did improve the performance. We will further investigate the reason behind this phenomenon to justify whether it is reasonable to add such a loss term. Thank you for your questions and inquiries about this.
>
> **Q4: This is not a proper choice, as there are many publicly available k-space datasets.**
> **Answer:**  We maintained the identical setting as our baseline to make the fair comparisons as possible, but only to find the sad fact that this setting is not in line with the mainstream. Your suggestion is good, and we will seek a better setting and revise this work accordingly. Thanks for your time and patience again.
>
> **Best Regards.**

---

### Official Review · Reviewer_KruX · 2024-11-04

**Soundness:** 2
**Presentation:** 1
**Contribution:** 2
**Rating:** 3
**Confidence:** 5

**Summary:**

This paper proposes a 3D Gaussian-based iterative reconstruction framework, termed Gaussian-Based Iterative Reconstruction (GBIR), to address the challenge of sparse sampling reconstruction in various medical imaging applications. Inspired by the 3D Gaussian splatting technique, this method reconsiders the practical demands of medical imaging, suggesting the abandonment of the splatting operation. Instead, it leverages 3D Gaussian representations to discretize the data into a volumetric form and applies an iterative reconstruction framework. Extensive experiments demonstrate that GBIR significantly outperforms other baseline methods. Additionally, the paper introduces a comprehensive Multi-Organ Medical Image REconstruction (MORE) dataset.

**Strengths:**

- This paper presents an efficient iterative reconstruction framework based on 3D Gaussian representation, introducing a set of strategies to enhance reconstruction efficiency. These include: 1) a 3D Gaussian representation truncated using the 3-sigma principle, and 2) efficient volume reconstruction through decomposition and parallelization. Experimental results demonstrate that these measures significantly reduce the spatial and temporal complexity of the reconstruction process.

- Additionally, this paper proposes a comprehensive multi-organ medical image reconstruction dataset, which, if open-sourced as anticipated, would provide substantial value to the research community.

**Weaknesses:**

- The paper presents an extensive comparison of quantitative results; however, it lacks any qualitative comparisons. In the field of medical image reconstruction, qualitative comparisons often provide a more accurate reflection of a method’s true performance. As the saying goes, “Visual results don’t lie.”

- Recently, several studies have proposed incorporating the principles of 3D Gaussian representation into medical image reconstruction [1-4]. There has also been some exploration of approaches that employ 3D Gaussian representation without using a splatting process [1,3]. This suggests that the proposed method may lack sufficient novelty. The authors should conduct a more comprehensive literature review in their paper and compare their method with SOTA 3D Gaussian-based techniques for medical image reconstruction.

> [1] Cai, Yuanhao, et al. "Radiative gaussian splatting for efficient x-ray novel view synthesis." European Conference on Computer Vision. Springer, Cham, 2025.

> [2] Zha, Ruyi, et al. "R $^ 2$-Gaussian: Rectifying Radiative Gaussian Splatting for Tomographic Reconstruction." arXiv preprint arXiv:2405.20693 (2024).

> [3] Fu, Xueming, et al. "3DGR-CAR: Coronary artery reconstruction from ultra-sparse 2D X-ray views with a 3D Gaussians representation." International Conference on Medical Image Computing and Computer-Assisted Intervention. Cham: Springer Nature Switzerland, 2024.

> [4] Li, Yingtai, et al. "Sparse-view ct reconstruction with 3d gaussian volumetric representation." arXiv preprint arXiv:2312.15676 (2023).

**Questions:**

- In line 161, the authors state that "DIFGaussian is based on 3D Gaussian splatting, whereas our proposed GBIR does not involve any ‘splatting’ process." However, to my knowledge, DIF-Gaussian also does not utilize a ‘splatting’ process, as discussed in detail in [MICCAI 2024 - Open Access Page](https://papers.miccai.org/miccai-2024/448-Paper0250.html).

- The authors conducted experiments on the compressed sensing MRI reconstruction task; however, they appear to have omitted specific details regarding the task setup, such as the exact acceleration factors and sampling patterns used.

---

> ### Author Response · Authors · 2024-11-26
>
> **Thank you sincerely for your time and valuable feedback. Below is our response to the reviewers' comments:**
>
> **Q1. The paper lacks qualitative comparisons.**
> **Answer:** We provided the visualization in Appendix Figure 5. We will include more qualitative comparisons in the revised manuscript to provide a more comprehensive evaluation of our proposed method.
>
> **Q2: The proposed method may lack novelty.**
> **Answer**: We respectfully disagree with this comment. Our proposed GBIR is not based on 3D Gaussian Splatting, which is different from the X-Gaussian [1], $R^2$ Gaussian [2], and 3DGR-CAR [3]. Instead of adapting 3DGS for medical image reconstruction, we propose a different 3D Gaussian representation and as well as an efficient iterative reconstruction framework. We will further clarify this in the revised manuscript.
>
> **Q3: Lack of details regarding the MRI task setup.**
> **Answer:** We uniformly sample the k-space data following previous work DiffusionMBIR [4], with a simple visualization shown in Figure 1 (IV part) the Mask image. We will revise the manuscript to provide more details on the MRI task setup.
>
> [1] Cai, Yuanhao, et al. "Radiative gaussian splatting for efficient x-ray novel view synthesis." European Conference on Computer Vision. Springer, Cham, 2025.
> [2] Zha, Ruyi, et al. "$R^2$-Gaussian: Rectifying Radiative Gaussian Splatting for Tomographic Reconstruction." arXiv preprint arXiv:2405.20693 (2024).
> [3] Fu, Xueming, et al. "3DGR-CAR: Coronary artery reconstruction from ultra-sparse 2D X-ray views with a 3D Gaussians representation." International Conference on Medical Image Computing and Computer-Assisted Intervention. Cham: Springer Nature Switzerland, 2024.
> [4] Hyungjin Chung, et al. "Solving 3d inverse problems using pre-trained 2d diffusion models." Conference on Computer Vision and Pattern Recognition, 2023.

---

> ### Comment · Reviewer_KruX · 2024-11-28
>
> Thank you for your prompt responses. However, my major concerns are not addressed. Firstly, this work aims to propose a novel method for medical inverse imaging, so visual comparisons between the proposed method and baselines are required. Currently, Fig. 5 in the Appendix only illustrates the reconstructions of the proposed method across iterations. Moreover, you mention that "Our proposed GBIR is not based on 3D Gaussian Splatting." However, the key formulas of the proposed GBIR (Eq. 3 and Eq. 4 in this submission) appear to align with the core concepts of R2-Gaussian based on 3D GS (Eq. 3 and Eq. 4 (https://arxiv.org/pdf/2405.20693)) in essence. Please provide further clarification on these points.

---

> ### Author Response · Authors · 2024-11-28
> **Visualization and Clarification on the Mathematical Basis of Our Work**
>
> Thank you very much for your additional feedback!
>
> **Volume Visualization:** We show the visualization of different methods at the [Anonymous Website](https://more-med.github.io/).
>
> **Clarification of GBIR:** We would like to clarify that our work is fundamentally based on the classical Gaussian distribution, introduced by Carl Friedrich Gauss in 1809, and not derived from 3D Gaussian Splatting (3DGS).
>
> 1. **Point 1: Our formula is based on Gauss's theory**
>    The **exponent term** of Gaussian distribution is given by:
>    $$G(\mathbf{x}, {\mu}, {\Sigma}) = \exp\left(-\frac{1}{2}(\mathbf{x} - {\mu})^\top {\Sigma}^{-1} (\mathbf{x} - {\mu})\right)$$
>    where $\mathbf{x} \in \mathbb{R}^d$ is a 3D point, $\mu$ is the mean, and $\Sigma$ is the covariance matrix. This formula is the Gaussian's exponent, a standard result in probability theory, not specific to 3DGS.
>
> 2. **Point 2: Our Reconstruction Target is Based on GMM**
>     We formulate the target volume as:
>    $$\mathbf{V} = \sum_{i=1}^n G(\mathbf{x}, {\mu}_i, {\Sigma}_i) \cdot I_i$$
>     where $I_i$ is the intensity of the $i$-th Gaussian, which is learnable. This formulation is a variant of the Gaussian Mixture Model (GMM). While GMMs use a sum of Gaussians with normalized weights that sum to one, our formulation uses learnable intensities $I_i$​ as weights. Thus, our reconstrcution target is a learnable-weight version of the GMM, which is a standard approach in machine learning for modeling distributions, and is not specific to methods like 3DGS or $R^2$ Gaussian.
>
> 3. **Point 3: Different Reconstruction and Optimization**
>     **Reconstruction**: Our method discretizes Gaussians into a 3D grid to directly reconstruct the target volume in an end-to-end manner. We maintain differentiability through **contribution alignment** and significantly accelerate reconstruction using **3-sigma truncation** and efficient **large Einstein summation decomposition**.
>
>     **Optimization**: We perform optimization in the **measurement domain** by transforming the reconstructed volume into the measurement space and minimizing the difference between the transformed volume and the measured data. This approach is fundamentally different from 3DGS, which optimizes by randomly rendering a view, then minimizing the difference between the rendered view and the ground truth image. $R^2$-Gaussian also optimizes in the image domain: and they render the X-ray projection and compute the loss with the ground truth image.
>
> **Best Regards.**

---

> > ### Comment · Reviewer_KruX · 2024-11-29
> >
> > Thank you for providing the visualization results. These results should be included in the main paper to improve clarity. However, I find the explanation regarding the novelty of GBIR unconvincing.  Below are my specific comments:
> >
> > 1.  3D GS is the pioneering work that uses a set of Gaussians to represent 3D scenes.  Many subsequent works have applied such Gaussian representations in medical imaging reconstruction.  The proposed method also uses Gaussian representations to resolve high-quality CT/MR images.  Thus, I do not believe this work introduces any innovation in signal representation.
> >
> > 2.  You mention "_We perform optimization in the measurement domain ...  This approach is fundamentally different from 3DGS, which optimizes by randomly rendering a view ...  R2-Gaussian also optimizes in the image domain:  and they render the X-ray projection and compute the loss with the ground truth image_".  These statements are quite confusing.  To my knowledge, in 3D GS the rendered images are measurement data, and for R2-Gaussian, the projection data is exactly the measurement data.  Therefore, the optimization framework of the proposed GBIR is generally the same as those of these previous works.

---

> ### Author Response · Authors · 2024-11-29
> **Thanks for Your Additional Comments**
>
> Thanks sincerely for your additional discussion and question. We will include the visualization in the revised manuscript.
>
> ### Innovation of scene representation:
> Yes, 3DGS is the pionner of using Gaussians in representing the scene, and we also stated that we were inspired by this excellent work in our **Introduction Section**. In the revised manuscript, we will further emphasize our innovative **reconstruction** and **optimization** processes.
>
> ### Optimizing in image domain or measurement domain?
> We are confident that both 3DGS and $R^2$-Gaussian optimize in the image domain. Below, we explain why.
>
> **For 3D Gaussian Splatting:**
> In 3DGS, the optimization process aligns rendered images with target images by minimizing their differences. During each iteration, a viewpoint is randomly selected, an image is rendered, and the loss is computed between the rendered and ground-truth images. This confirms that the optimization occurs in the image domain.
>
> We show how they optimize in the image domain below. ([Link of Code](https://github.com/graphdeco-inria/gaussian-splatting/blob/main/train.py))
>
> **Select the viewpoint, and render the image from Gaussians**
> ```
> render_pkg = render(viewpoint_cam, gaussians, pipe, bg, use_trained_exp=dataset.train_test_exp, separate_sh=SPARSE_ADAM_AVAILABLE)
> ```
>
> **Get the rendered image**
> ```
> image, viewspace_point_tensor, visibility_filter, radii = render_pkg["render"], render_pkg["viewspace_points"], render_pkg["visibility_filter"], render_pkg["radii"]
> if viewpoint_cam.alpha_mask is not None:
>     alpha_mask = viewpoint_cam.alpha_mask.cuda()
>     image *= alpha_mask
> ```
> **Then the loss is computed with the rendered image and the ground truth image, which is in the image domain.**
> ```
> gt_image = viewpoint_cam.original_image.cuda()
> Ll1 = l1_loss(image, gt_image) #
> if FUSED_SSIM_AVAILABLE:
>     ssim_value = fused_ssim(image.unsqueeze(0), gt_image.unsqueeze(0))
> else:
>     ssim_value = ssim(image, gt_image)
> loss = (1.0 - opt.lambda_dssim) * Ll1 + opt.lambda_dssim * (1.0 - ssim_value)
> ```
>
> **For the $R^2$-Gaussian:**
> In $R^2$-Gaussian, they optimize the loss with rendered projection and measured projection in Formula 8. Below is their implementation and we explain how they optimize in the image domain below.
> ([Link of Code](https://github.com/Ruyi-Zha/r2_gaussian/blob/main/train.py))
>
> **Select a viewpoint and render projection data $I_r$ in this selected viewpoint**
> ```
> # Get one camera for training
> if not viewpoint_stack:
>     viewpoint_stack = scene.getTrainCameras().copy()
> viewpoint_cam = viewpoint_stack.pop(randint(0, len(viewpoint_stack) - 1)) # Similar to 3DGS to select a viewpoint
> render_pkg = render(viewpoint_cam, gaussians, pipe) # Render X-ray projection in this selected viewpoint
> ```
>
> **Get the rendered projection data $I_r$**
> ```
> image, viewspace_point_tensor, visibility_filter, radii = (
>     render_pkg["render"],
>     render_pkg["viewspace_points"],
>     render_pkg["visibility_filter"],
>     render_pkg["radii"],
> )
> ```
>
> **Then Compute the loss between rendered projection $I_r$ and ground truth projection $I_m$, both are X-ray images instead of measurement. Note that there is not any Radon Transform involved in transforming the image domain into measurement domain.**
> ```
> gt_image = viewpoint_cam.original_image.cuda()
> loss = {"total": 0.0}
> render_loss = l1_loss(image, gt_image)
> ```
> **Thus, we are sure that the 3DGS and $R^2$-Gaussian are optimized in the image domain instead of measurement domain, and they do not involve any Radon Transform to project the image domain into measurement domain.**
>
> **We also show how our GBIR works:**
> On the contrary, our GBIR directly optimize under the measurement domain, i.e., without the need of image for supervision.  We show how we optimize in the measurement domain below.
>
> **Directly reconstruct the volume from the Gaussians Set，without any rendering process.**
> ```
> reconstruct_volume, impact_shape, std = reconstruct_3d_volume(gaussians, gaussians.volume_shape)
> ```
> **Project the volume into measurement domain**
> ```
> projs = gaussians.Fan_ray_trafo(reconstruct_volume)
> ```
> **Compute the L1-Loss in the measurement domain**
> ```
> Ll1 = l1_loss(projs, gaussians.gt_projs)
> ```
>
>
>
> As shown above, our **Reconstruction** as well as **Optimization** are innovative: we did not involve any viewpoint selection / rendering process in the reconstruction. Instead, we propose a highly efficient reconstruction directly from a set of Gaussians; and our optimization is completely done in the measurement domain, without any supervision signal in the image domain.

---

> > ### Comment · Reviewer_KruX · 2024-12-01
> >
> > Thank you for your response. Unfortunately, in my humble opinion, your statement "3D GS and R2-Gaussian are optimized in the image domain instead of the measurement domain" appears to be incorrect. Here are my specific comments: 1) 3D GS learns Gaussian representations of 3D scenes from limited-view 2D images. The optimization goal of 3D GS is to minimize errors between estimated and measured 2D images. This suggests that 3D GS is optimized in the measurement domain; 2) R2-Guassian aims to address sparse-view CT reconstruction in an unsupervised manner, which is also preserved by GBIR. R2-Guassian can recover unknown 3D CT volumes from sparse-view 2D projections. In the context of CT reconstruction, these 2D projections are the measurement data. Your statement that "they do not involve any Radon transform to project the image domain into the measurement domain" is quite confusing. R2-Gaussian is an unsupervised method and does not use any image domain information. Instead, it uses a ray-based integral model (Eq. 5) to generate projections (i.e., measurement data). This integral model is equivalent to Radon transformation in essence. Furthermore, its loss function (Eq. 8) is explicitly designed based on the error between the real and estimated projections, which is rooted in the measurement domain. Moreover, I would like to emphasize that under the context of 3D CT reconstruction, the 2D X-ray projections are exactly measurement data and cannot be viewed as 2D images.

---

> > > ### Author Response · Authors · 2024-12-04
> > > **Clarification on the Term "Measurement" in Medical Image Reconstruction**
> > >
> > > **Dear Reviewer KruX,**
> > > Thank you very much for your polite and continuous feedback. We would like to clarify the different interpretations of the term **measurement** in the context of medical image reconstruction, which we believe has led to some differences in understanding the methods.
> > >
> > > In medical image reconstruction, the term **measurement** typically refers to the raw data [1][2] obtained directly from the imaging sensors or detectors during the scanning process. For example, in **CT**, the measurement data consists of the **sinogram**, which represents 2D projections collected from multiple angles during the scan. In **MRI**, it refers to the data in **k-space**, which is the raw frequency-domain data that is later transformed into spatial images. These raw data sets are essential for the reconstruction process and form the basis for generating the final images.
> > >
> > > However, when referring to **measured 2D images,** it is important to note that this is not the typical understanding of "measurement" in this field. In **3DGS**, the measured images are optical images captured from specific views, which differs from the raw measurement data. In **$R^2$-Gaussian**, the measured projections are 2D images that represent the body’s internal structures, not the raw "measurement" data itself. A comparison of our CT measurement (Figure 2, bottom-right) with $R^2$-Gaussian's projection (Figure 7 in their work) shows that their projection contains internal structures, indicating it belongs to the image domain rather than the raw data domain. In contrast, our measurement in the figure depicts the CT sinogram, which is the true measurement data.
> > >
> > > **Thank you again for your valuable feedback and continued discussion.**
> > >
> > > **Best regards.**
> > >
> > > [[1] Lennart Koetzier et al. Deep learning image reconstruction for ct: Technical principles and clinical prospects.](https://pubs.rsna.org/doi/full/10.1148/radiol.221257)
> > > [[2] Florian Knoll et al.  fastmri: A publicly available raw k-space and dicom dataset of knee images for accelerated mr image reconstruction using machine learning.](https://pmc.ncbi.nlm.nih.gov/articles/PMC6996599/)

---

### Author Response · Authors · 2024-11-28
**Paper Revision in Response to Reviewer Feedback**

We sincerely thank all the reviewers for their valuable feedback and constructive suggestions. In response, we have carefully revised the manuscript to address the concerns, and we include the 3d visualization comparison in this [Anonymous Website](https://more-med.github.io/).

Below, we highlight the key changes:

1. **Relationship to Contemporary Research**:
   We have added a new paragraph, **Relationship with Existing Works**, in Section 2 (**Related Work**) to discuss how our work differs from contemporary 3DGS-Based medical image reconstruction methods.

2. **Clarification of the MRI Setting**:
   We clarified the simulation setting in Section 5.1 (**Experimental Settings**) and provided additional details about the computation of the loss function of complex-valued $k$-space measurement in Section 3.3 (**Optimization in Measurement Domain**).

3. **Description of the GBIR Method**:
   We specified that the Gaussian functions used in the GBIR method are isotropic in Section 3.1 and included the hyperparameters of the loss function in Section 3.3.

4. **Description of the MORE Dataset**:
   We expanded Section 4 to provide a detailed description of the MORE dataset, including the data provided and the simulation process.

To comply with the page limit, we adjusted line spacing and moved the original Table 1 (**Different Types of Medical Image Reconstruction Methods**) to the Appendix (now Table 7). Additionally, we revised and polished the manuscript to correct writing errors and typos.

**Best Regards,**
The Authors

---

### Note · Authors · 2024-12-11

**Comment:**

We sincerely thank all reviewers and readers for their time, effort, and insightful feedback on our research. After thorough discussions and careful consideration, we have decided to withdraw our submission.

In this work, we proposed a novel end-to-end framework for medical image reconstruction that diverges from 3DGS-based methods, alongside introducing the Multi-Organ Reconstruction (MORE) dataset and a comprehensive benchmark. While these contributions offer potential value to the field, the reviewers rightly highlighted certain limitations, including the divergence of our MRI settings from mainstream practices and the absence of comparisons with some relevant contemporaneous methods.

In light of this constructive feedback and the valuable discussions we had with the reviewers, we have chosen to withdraw the paper to refine and enhance it further.

To those interested in our research, we encourage you to stay tuned for an updated version. We remain confident that our methods, dataset, and benchmark will make a meaningful contribution to advancing medical image reconstruction in the near future.

**Sincerely,**
The Authors

**Withdrawal Confirmation:**

I have read and agree with the venue's withdrawal policy on behalf of myself and my co-authors.